# Bilevel ZOFO: Efficient LLM Fine-Tuning and Meta-Training

**Reza Shirkavand**
Department of Computer Science
University of Maryland - College Park
rezashkv@cs.umd.edu

**Peiran Yu**
Department of Computer Science and Engineering
University of Texas at Arlington
peiran.yu@uta.edu

**Qi He**
Department of Computer Science
University of Maryland - College Park
qhe123@cs.umd.edu

**Heng Huang**[*]
Department of Computer Science
University of Maryland - College Park
heng@cs.umd.edu

## Abstract

Fine-tuning pre-trained Large Language Models (LLMs) for downstream tasks using First-Order (FO) optimizers presents significant computational challenges. Parameter-Efficient Fine-Tuning (PEFT) methods address these by freezing most model parameters and training only a small subset. However, PEFT often underperforms compared to full fine-tuning when high task-specific accuracy is required. Zeroth-Order (ZO) methods fine-tune the entire pre-trained model without back-propagation, estimating gradients through forward passes only. While memory-efficient, ZO methods suffer from slow convergence and high sensitivity to prompt selection. We bridge these two worlds with Bilevel-ZOFO, a bilevel optimization method that couples fast, local FO-PEFT adaptation at the inner level with stable, memory-efficient ZO updates of the full backbone at the outer level. The FO-PEFT inner loop performs fast, low-memory local adaptation that reduces the variance of ZO estimates and stabilizes the search, guiding the outer ZO updates of the full backbone and reducing prompt sensitivity. In the mean time, the outer ZO provides better generalization ability for PEFT. We provide theoretical convergence guarantees and empirically demonstrate that Bilevel-ZOFO significantly outperforms existing ZO and FO-PEFT methods, achieving 2–4× faster training while maintaining similar memory efficiency. Additionally, we show by updating the backbone with ZO and adapting only a tiny FO-PEFT block per task, Bilevel-ZOFO combines full-model capacity with few-shot efficiency, making it a very efficient meta-learning algorithm that quickly adapts to new tasks.

## 1 Introduction

Fine-tuning pretrained Large Language Models (LLMs) has become a standard approach for downstream tasks. Traditional First-Order (FO) optimizers like Adam [20], commonly used for this process, rely on backpropagation. However, as highlighted in Malladi et al. [35], computing gradients for LLMs can require up to 12 times the memory needed for inference. This scaling challenge becomes even more pronounced as models grow larger, imposing significant memory demands and complicating the fine-tuning process, especially in resource-constrained environments.

---

[*]This work was partially supported by NSF IIS 2347592, 2348169, DBI 2405416, CCF 2348306, CNS 2347617, RISE 2536663.

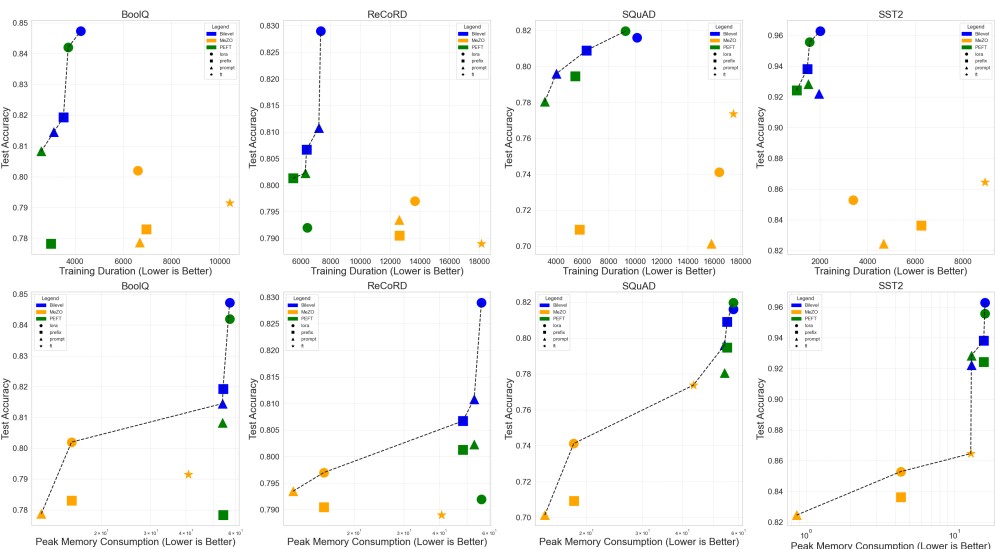

Figure 1: **LLaMA-7B: (top) Accuracy vs. training duration and (bottom) Accuracy vs. memory across four tasks.** Almost all *bilevel-zofo* points lie on the Pareto frontier and higher than the baselines. **(top)** Our method (blue) achieves significantly higher accuracy than MeZO (orange) while being ~2–4× faster in training offering a better trade-offs in accuracy and runtime. **(bottom)** Bilevel-zofo maintains the same memory footprint and significantly outperforms each corresponding PEFT variant (compare blue vs. green circles, squares, and triangles). *Bilevel-zofo* effectively combines the efficiency and expressivity of full ZO fine-tuning with the speed and robustness of first-order methods. It incorporates elements from both baselines but takes a definitive step beyond them.

To address these computational challenges, Parameter-Efficient Fine-Tuning (PEFT) methods have been developed. These techniques freeze most of the model's parameters and train only a small subset, significantly reducing both memory and computational overhead. Popular PEFT approaches include prompt tuning, LoRA fine-tuning, and prefix tuning. Prompt tuning [22, 40, 58, 48] optimizes continuous prompt vectors that are concatenated with the input embeddings, while prefix tuning [24] introduces learnable prefix tokens that serve as conditioning variables at each transformer layer. LoRA (Low-Rank Adaptation) [16, 15] modifies the model's attention and feedforward layers by injecting low-rank trainable matrices, further reducing the resources required for fine-tuning.

While Parameter-Efficient Fine-Tuning (PEFT) methods reduce training costs and memory usage, they may not always achieve the same level of task-specific performance as full model fine-tuning. Research has shown that for tasks requiring high accuracy, complex adaptations, or domain-specific knowledge, full fine-tuning often outperforms PEFT approaches due to its ability to adjust all model parameters for better adaptation [16, 24, 59]. To make full model fine-tuning more computationally feasible, Zeroth-Order (ZO) methods offer an alternative by reducing the high computational cost. Rather than computing gradients via backpropagation, zeroth-order methods estimate the gradient using only the forward pass. Initially explored in the 1990s [50, 38, 11, 6, 30], these methods have recently gained traction for fine-tuning LLMs [35, 4, 25] and have been shown to be able to outperform FO PEFT methods given enough training time [63].

A major limitation of ZO methods is their slow convergence due to the need for gradient estimation. For instance, MeZO [35] required 10 times more iterations than PEFT baselines to match or exceed their performance. Additionally, ZO methods suffer from extreme sensitivity to prompt selection. In tasks like sentiment analysis with the SST-2 dataset, templated prompts (e.g., *"$<CLS>$ text data. It was [terrible | great]. $<SEP>$"*) are crucial for success [63]. These prompts effectively align the text data with task-specific objectives. As a result, prompt selection becomes an important hyperparameter that can significantly affect performance. In particular, ZO methods have been shown to be highly sensitive to prompt selections [35]. Without proper prompts, the performance of MeZO can drastically drop (Table 1).

In this paper, we ask: Can zeroth-order (ZO) and PEFT methods be smoothly integrated to mutually enhance each other—achieving greater robustness to prompt variations, faster convergence, and better

performance than either method alone—while maintaining memory efficiency comparable to each individually? We target settings where (i) full FO fine-tuning is impractical due to memory/throughput, (ii) pure PEFT lacks full-model capacity on harder adaptations, and (iii) pure ZO is slow and highly prompt-sensitive.

We propose Bilevel-ZOFO, a novel bilevel optimization framework explicitly designed to leverage the complementary strengths of these two approaches:

- At the inner level, FO-PEFT rapidly performs targeted, local adaptation using first-order gradients, stabilizing training and mitigating sensitivity to task-specific prompts that ZO methods need.

- At the outer level, a ZO method updates the full backbone model parameters efficiently, guided by the stable and informative inner-level adaptation. This full model finetuning enhances the model's generalization ability, enables a more sophisticated understanding, and improves transfer to new tasks.

This clear separation enables efficient bilevel optimization, addressing the major drawbacks of pure ZO methods (slow convergence, prompt sensitivity) and pure PEFT methods (limited full-model adaptation). Extensive ablation studies empirically verify this synergy, demonstrating faster convergence and more robust performance.

## 1.1 Contributions

We summarize our main contributions as follows:

1. We propose Bilevel-ZOFO, a theoretically grounded and practical bilevel optimization method that enhances zeroth-order (ZO) optimization by selecting the best prompt and thereby improves ZO fine-tuning with first-order PEFT (FO-PEFT). At the same time, it strengthens PEFT by leveraging full fine-tuning through ZO updates.

2. Bilevel-ZOFO reduces ZO sensitivity to prompt choices and significantly accelerates convergence, achieving state-of-the-art performance with minimal memory overhead.

3. Extensive experiments confirm that Bilevel-ZOFO consistently outperforms existing FO-PEFT and ZO baselines across diverse tasks.

4. By updating the backbone with ZO and adapting only a tiny FO-PEFT block per task, our method couples full-capacity transfer with few-shot efficiency. We show that this design has strong potential for efficient meta-learning, demonstrating improved multi-task adaptation with minimal computational resources.

## 2 Related work

### 2.1 Zeroth-Order Methods in Fine-Tuning LLMs

MeZO [35] pioneered zeroth-order (ZO) fine-tuning for LLMs, demonstrating compatibility with both full-model and PEFT approaches while improving computational efficiency. Subsequent work provided a benchmark for ZO optimization methods [63], and expanded ZO applications to variance reduction [10], federated fine-tuning [41, 25], softmax layers [5], sparse tuning [14, 31], and privacy [51]. In contrast, we propose a bilevel training framework that unifies the strengths of ZO full-model and FO PEFT fine-tuning, outperforming both individually, while using as much resources.

### 2.2 First-Order Methods for Bilevel Optimization

Bilevel optimization is computationally demanding, especially for LLMs, due to the cost of computing hypergradients. Classical approaches rely on second-order methods [8, 9, 7, 23, 43, 12, 2, 32], while recent work [33, 46, 29, 21, 26, 18, 34] bypass the need for second-order information by reformulating the bilevel problem as a constrained optimization problem. We build on this by incorporating ZO approximations into the upper-level optimization to bypass full gradient computation for LLMs. These methods significantly reduce computational costs by eliminating the need for second-order information. Nevertheless, when fine tuning LLMs, back propagation for calculating the gradient

of an LLM is still too expensive. Liu et al. [29] and Lu and Mei [33] explore the convergence of their proposed methods to the original bilevel problem, while other approaches only demonstrate convergence to the penalized problem. In this paper, we adapt the method from Lu and Mei [33] to approximate part of the upper-level parameters using a ZO approximation in order to address the challenge posed by the large number of training parameters in large language models. We also provide convergence guarantees for this adapted zeroth-order-first-order method.

# 3 Bilevel Model with Zeroth-Order-First-Order Method

In this section, we introduce our bilevel model and the zeroth-order-first-order method for solving it.

## 3.1 Preliminaries and Notation

Let $\mathbf{p} \in \mathbb{R}^{d'}$ represent the parameters of the PEFT model, and $\boldsymbol{\theta} \in \mathbb{R}^d$ represent the parameters of the pretrained base model. We denote the loss function given a dataset $\mathcal{D}$ as $F(\boldsymbol{\theta}, \mathbf{p}; \mathcal{D}) := \frac{1}{|\mathcal{D}|} \sum_{x \in \mathcal{D}} F(\boldsymbol{\theta}, \mathbf{p}; x)$. Given a single downstream task, such as classification, we aim to solve the following optimization problem:

$$\min_{\boldsymbol{\theta} \in \mathbb{R}^d} F(\boldsymbol{\theta}, \mathbf{p}; \mathfrak{D}). \tag{1}$$

Where $\mathbf{p}$ corresponds to the embeddings of the hard prompt (as shown in Table 13 in the appendix of [35]), the model above reduces to classical fine-tuning on a single downstream task. In model (1), the parameters of the PEFT model, $\mathbf{p}$, are fixed.

To enhance generalization ability, we split the dataset $\mathfrak{D}$ into two parts: one for tuning the PEFT model (denoted by $\mathfrak{D}_{\mathbf{p}}$) and another for fine-tuning the LLM (denoted by $\mathfrak{D}_f$). To maximize performance on downstream tasks, we need the optimal PEFT model parameters that are best suited for the current LLM base model. To achieve this, we require $\mathbf{p}$ to satisfy the following condition:

$$\mathbf{p} \in \arg\min_{\mathbf{s} \in \mathbb{R}^{d'}} F(\boldsymbol{\theta}, \mathbf{s}; \mathfrak{D}_{\mathbf{p}}).$$

Here $s$ is just the dummy optimization variable for the inner problem—i.e., a candidate PEFT-parameter vector over which we minimize to obtain the optimal $p$ for the current $\theta$ on $\mathfrak{D}_{\mathbf{p}}$. [2] This condition reveals that as the parameters $\theta$ of the LLM change, the parameters $\mathbf{p}$ in the PEFT model should also be updated accordingly to be the best match for $\theta$. Therefore, instead of solving (1), our true objective becomes:

$$\min_{\boldsymbol{\theta} \in \mathbb{R}^d} F(\boldsymbol{\theta}, \mathbf{p}; \mathfrak{D}_f) \text{ s. t. } \mathbf{p} \in \arg\min_{\mathbf{s} \in \mathbb{R}^{d'}} F(\boldsymbol{\theta}, \mathbf{s}; \mathfrak{D}_{\mathbf{p}}). \tag{2}$$

In this way, we find the optimal pair of parameters for both the PEFT model and the LLM base model to achieve the best performance on downstream tasks.

## 3.2 Bilevel Model

Eq. (2) is an instance of a bilevel optimization problem. To solve it, classical bilevel methods (as discussed in related work) view Eq. (2) as a single-level problem $\min_{\boldsymbol{\theta}} F(\boldsymbol{\theta}, \mathbf{p})$. Since $\mathbf{p}$ is the minimizer of another optimization problem, these methods typically require computing the Hessian-vector product (matrix multiplication of $\nabla_{\boldsymbol{\theta}\mathbf{p}} F(\boldsymbol{\theta}, \mathbf{p})$ and some vector $v$) multiple times to estimate the gradient of $F(\boldsymbol{\theta}, \mathbf{p})$ with respect to $\boldsymbol{\theta}$. However, for large language models (LLMs), this approach is computationally prohibitive because the number of parameters in $\boldsymbol{\theta}$ is too large.

To reduce the computational cost, following [33], we consider using a penalty method for the bilevel problem (2). Specifically, (2) is equivalent to the following constrained optimization problem:

$$\min_{\boldsymbol{\theta} \in \mathbb{R}^d, \mathbf{p} \in \mathbb{R}^{d'}} F(\boldsymbol{\theta}, \mathbf{p}; \mathfrak{D}_f) \text{ s.t. } F(\boldsymbol{\theta}, \mathbf{p}; \mathfrak{D}_{\mathbf{p}}) - \inf_{\mathbf{s}} F(\boldsymbol{\theta}, \mathbf{s}; \mathfrak{D}_{\mathbf{p}}) \leq 0. \tag{3}$$

By penalizing the constraint with a constant $\lambda > 0$, we obtain the following penalized problem:

$$\min_{\substack{\boldsymbol{\theta} \in \mathbb{R}^d \\ \mathbf{p} \in \mathbb{R}^{d'}}} F(\boldsymbol{\theta}, \mathbf{p}(\boldsymbol{\theta}); \mathfrak{D}_f) + \lambda(F(\boldsymbol{\theta}, \mathbf{p}; \mathfrak{D}_{\mathbf{p}}) - \inf_{\mathbf{s} \in \mathbb{R}^{d'}} F(\boldsymbol{\theta}, \mathbf{s}; \mathfrak{D}_{\mathbf{p}})). \tag{4}$$

---

[2] $\mathbf{s}$ does not introduce new parameters. It only denotes the search variable of the inner minimization.

---

**Algorithm 1** Bilevel first-order method

---

1: **Symbols:** $\boldsymbol{\theta} \in \mathbb{R}^d$ (backbone params), $\mathbf{p} \in \mathbb{R}^{d'}$ (PEFT params), $\mathbf{s} \in \mathbb{R}^{d'}$ (aux inner variable), $G_\lambda(\boldsymbol{\theta}, \mathbf{p}, \mathbf{s})$ (penalized objective), $K$ (number of outer steps), $T$ (number of inner steps between two outer steps), $\eta > 0$ (inner LR), $\zeta > 0$ (outer LR), $\{\lambda_k\} \subseteq \mathbb{R}_+$ (penalty at step $k$).
2: **Input:** step sizes $\eta, \zeta > 0$; initial states $\boldsymbol{\theta}^0, \mathbf{p}^0, \mathbf{s}^0$; $K, T \in \mathbb{N}_+$; penalty schedule $\{\lambda_k\}_{k=0}^{K-1}$.
3: **for** $k = 0, \ldots, K-1$ **do**
4:     **for** $t = 0, \ldots, T-1$ **do**
5:         $\mathbf{s}_{t+1}^k = \mathbf{s}_t^k - \eta \, \nabla_{\mathbf{s}} \, G_{\lambda_k}(\boldsymbol{\theta}^k, \mathbf{p}^k, \mathbf{s}_t^k)$
6:     **end for**
7:     $\mathbf{s}^{k+1} \leftarrow \mathbf{s}_T^k$
8:     $\boldsymbol{\theta}^{k+1} = \boldsymbol{\theta}^k - \zeta \, \nabla_{\boldsymbol{\theta}} \, G_{\lambda_k}(\boldsymbol{\theta}^k, \mathbf{p}^k, \mathbf{s}^{k+1})$
9:     $\mathbf{p}^{k+1} = \mathbf{p}^k - \zeta \, \nabla_{\mathbf{p}} \, G_{\lambda_k}(\boldsymbol{\theta}^k, \mathbf{p}^k, \mathbf{s}^{k+1})$
10: **end for**

---

As $\lambda$ increases, the solution to the penalized problem approaches the solution to (3), and thus the solution to (2) (see Lemma B.4 for an explicit relationship between the stationary points of (4) and those of the original problem (2)). Note that the penalized problem (4) is equivalent to the following minimax problem:

$$\min_{\boldsymbol{\theta} \in \mathbb{R}^d, \mathbf{p} \in \mathbb{R}^{d'}} \max_{\mathbf{s} \in \mathbb{R}^{d'}} G_\lambda(\boldsymbol{\theta}, \mathbf{p}, \mathbf{s}) \quad := \quad F(\boldsymbol{\theta}, \mathbf{p}(\boldsymbol{\theta}); \mathfrak{D}_f) + \lambda\big(F(\boldsymbol{\theta}, \mathbf{p}; \mathfrak{D}_{\mathbf{p}}) - F(\boldsymbol{\theta}, \mathbf{s}; \mathfrak{D}_{\mathbf{p}})\big). \quad (5)$$

In this way, we can solve the bilevel problem as a minimax problem. The basic minimax algorithm works as follows: at iteration $k$, we first solve the maximization problem $\max_{\mathbf{s}} G_\lambda(\boldsymbol{\theta}^k, \mathbf{p}^k, \mathbf{s})$ with $(\boldsymbol{\theta}^k, \mathbf{p}^k)$ fixed. For example, we can update $\mathbf{s}^k$ using an inner loop with stochastic gradient descent (SGD). Let $\mathbf{s}^{k+1}$ be the result of this inner loop. Then, in the outer loop, we update $(\boldsymbol{\theta}^k, \mathbf{p}^k)$ by solving $\min_{\boldsymbol{\theta}, \mathbf{p}} G_\lambda(\boldsymbol{\theta}, \mathbf{p}, \mathbf{s}^{k+1})$ with $\mathbf{s}^{k+1}$ fixed. Again, SGD can be used to update $\boldsymbol{\theta}^k$ and $\mathbf{p}^k$. The conceptual algorithm is presented in Algorithm 1. We assume we do a total of $K$ outer iterations and $T$ inner iterations between each two consecutive outer steps.

However, note that

$$\nabla_{\boldsymbol{\theta}} G_{\lambda_k}(\boldsymbol{\theta}^k, \mathbf{p}^k, \mathbf{s}^k) = \nabla_{\boldsymbol{\theta}} F(\boldsymbol{\theta}^k, \mathbf{p}^k; \mathfrak{D}_f) + \lambda_k(\nabla_{\boldsymbol{\theta}} F(\boldsymbol{\theta}^k, \mathbf{p}^k; \mathfrak{D}_{\mathbf{p}}) + \nabla_{\boldsymbol{\theta}} F(\boldsymbol{\theta}^k, \mathbf{s}^k; \mathfrak{D}_{\mathbf{p}})), \quad (6)$$

requires calculating the gradient with respect to $\boldsymbol{\theta}$, i.e, $\nabla_{\boldsymbol{\theta}} F(\boldsymbol{\theta}^k, \mathbf{p}^k; \mathfrak{D}_f)$. Given the large scale of $\boldsymbol{\theta}$ in LLMs, this is computationally expensive. To avoid this, we use zeroth-order (ZO) information to approximate the gradient $\nabla_{\boldsymbol{\theta}} G$. Following [35, 63, 14], we employ the Simultaneous Perturbation Stochastic Approximation (SPSA) as a classical zeroth-order gradient estimator. Specifically, at each iteration $k$, we sample $\mathbf{z}^k \sim N(0, I_d)$, recalling that $d$ is the dimension of $\boldsymbol{\theta}$. We then approximate the gradient $\nabla_{\boldsymbol{\theta}} F$ as follows:

$$\hat{\nabla}_{\boldsymbol{\theta}} F(\boldsymbol{\theta}^k, \mathbf{p}^k; x) := \frac{F(\boldsymbol{\theta}^k + \epsilon \mathbf{z}^k, \mathbf{p}^k; x) - F(\boldsymbol{\theta}^k - \epsilon \mathbf{z}^k, \mathbf{p}^k; x)}{2\epsilon} \mathbf{z}^k. \quad (7)$$

As opposed to the number of LLM parameters $\boldsymbol{\theta}$, the number of PEFT parameters $\mathbf{p}$ is very small. So it is feasible to compute the exact gradient with respect to $\mathbf{p}$. Thus, we calculate $\nabla_{\mathbf{p}} F(\boldsymbol{\theta}, \mathbf{p}; \mathcal{B})$ exactly.

Additionally, in each iteration $k$, we sample mini-batches $\mathcal{B}_f^k$ and $\mathcal{B}_p^k$ and use $\hat{\nabla}_{\boldsymbol{\theta}} F(\boldsymbol{\theta}^k, \mathbf{p}^k; \mathcal{B})$ to substitute $\nabla_{\boldsymbol{\theta}} F(\boldsymbol{\theta}^k, \mathbf{p}^k; \mathcal{D}_f)$ and $\nabla_{\boldsymbol{\theta}} F(\boldsymbol{\theta}^k, \mathbf{p}^k; \mathcal{D}_p)$ in (6). We also use mini-batches when calculating the gradients with respect to the PEFT parameters $\mathbf{s}$ and $\mathbf{p}$.

This approach leads to the final algorithm (Algorithm 2 and Figure 5) for fine-tuning LLMs using the bilevel model (2). We refer to this method as the Bilevel Zeroth-Order-First-Order (Bilevel ZOFO) method. In Appendix B, we show that Bilevel ZOFO converges at a rate of $O(\epsilon^{-2})$ under widely accepted assumptions.[3] The complexity of Bilevel ZOFO matches that in previous ZO

---

[3] In our theorem, we assume strong convexity on the lower level objective function, as is common in other theoretical work on bilevel optimization. While strong convexity facilitates theoretical analysis, our experiments demonstrate that the method remains robust even when this condition is not strictly satisfied.

**Algorithm 2** Bilevel Zeroth-order-first-order Method (Bilevel ZOFO)

1: **Symbols:** $\boldsymbol{\theta} \in \mathbb{R}^d$ (backbone params), $\mathbf{p} \in \mathbb{R}^{d'}$ (PEFT params), $\mathbf{s} \in \mathbb{R}^{d'}$ (aux inner variable), $F(\boldsymbol{\theta}, \mathbf{p}; \mathcal{D})$ (avg. loss over data $\mathcal{D}$), $\widehat{\nabla}_{\boldsymbol{\theta}} F$ (ZO grad. estimator; see (7)), $\mathfrak{D}_{\mathbf{p}}$ (inner dataset), $\mathfrak{D}_f$ (outer dataset), $B$ (mini-batch size), $K$ (number of outer steps), $T$ (number of inner steps between two outer steps), $\eta > 0$ (inner LR), $\zeta > 0$ (outer LR), $\{\lambda_k\} \subseteq \mathbb{R}_+$ (penalty schedule).
2: **Input:** step sizes $\eta, \zeta > 0$; batch size $B$; datasets $\mathfrak{D}_{\mathbf{p}}, \mathfrak{D}_f$; initial states $\boldsymbol{\theta}^0, \mathbf{p}^0, \mathbf{s}^0$; $K, T \in \mathbb{N}_+$; penalty $\{\lambda_k\}_{k=0}^{K-1}$.
3: **for** k=0,...,K **do**
4:     **for** t=0,...,T-1 **do**
5:         Sample a batch $\mathcal{B}_{t,\mathbf{p}}^k$ from $\mathfrak{D}_{\mathbf{p}}$.
6:         Let $\mathbf{s}_{t+1}^k = \mathbf{s}_t^k - \eta \nabla_{\mathbf{s}} F(\boldsymbol{\theta}^k, \mathbf{s}_t^k; \mathcal{B}_{t,\mathbf{p}}^k)$
7:         Output $\mathbf{s}^{k+1} = \mathbf{s}_T^k$.
8:     **end for**
9:     Sample a batch $\{\mathcal{B}_f^k\}$ from $\mathfrak{D}_f$ and $\{\mathcal{B}_{\mathbf{p}}^k\}$ from $\mathfrak{D}_{\mathbf{p}}$.
10:    For $x \in \mathcal{B}_{\mathbf{p}}^k \cup \mathcal{B}_f^k$, calculate $\hat{\nabla}_{\boldsymbol{\theta}} F(\boldsymbol{\theta}^k, \mathbf{p}^k; x)$ following (7).
11:    Let

$$\mathbf{p}^{k+1} = \mathbf{p}^k - \zeta(\nabla_{\mathbf{p}} F(\boldsymbol{\theta}^k, \mathbf{p}^k; \mathcal{B}_f^k) + \lambda_k (\nabla_{\mathbf{p}} F(\boldsymbol{\theta}^k, \mathbf{p}^k; \mathcal{B}_{\mathbf{p}}^k))) \tag{8}$$

$$\boldsymbol{\theta}^{k+1} = \boldsymbol{\theta}^k - \zeta(\hat{\nabla}_{\boldsymbol{\theta}} F(\boldsymbol{\theta}^k, \mathbf{p}^k; \mathcal{B}_f^k) + \lambda_k (\hat{\nabla}_{\boldsymbol{\theta}} F(\boldsymbol{\theta}^k, \mathbf{p}^k; \mathcal{B}_{\mathbf{p}}^k) - \hat{\nabla}_{\boldsymbol{\theta}} F(\boldsymbol{\theta}^k, \mathbf{s}^{k+1}; \mathcal{B}_{\mathbf{p}}^k))) \tag{9}$$

12: **end for**

minimax algorithm in [55] but solves our bilevel optimization problem (2) and does not depend on the dimensionality $d$ thanks to the efficient rank assumption B.5, providing efficiency guarantee for our algorithm.

## 4 Experiments

We conduct extensive experiments on various LLMs of different scales to demonstrate the effectiveness of bilevel-ZOFO in improving current zeroth order methods and PEFT. We also conduct experiments in testing its potential in meta training. Noticing that our proposed structure is able to incorporate any variation of zeroth-order methods in the upper-level step and any PEFT method in the lower level, to maintain focus on testing the effectiveness of the proposed bilevel structure and its unique multitask learning capabilities, we used the classic MeZO [35].

### 4.1 Single Task Experiments

#### 4.1.1 Experimental Setting

Following MeZO [35], we evaluate our approach on a range of classification and multiple-choice tasks. In this setting, training and testing are conducted on the same task. We employ prompt-tuning [22], prefix-tuning [24], and LoRA [16]- well-known PEFT baselines-for lower-level training to validate bilevel-ZOFO under different conditions and resource constraints. During each lower-level update, we update only the PEFT parameters, and during the upper-level optimization step, we tune the full model using zeroth-order gradient approximation. We perform 10 lower-level updates between each pair of upper-level updates. For each task, we randomly sample 1000 examples for training, 500 examples for validation,

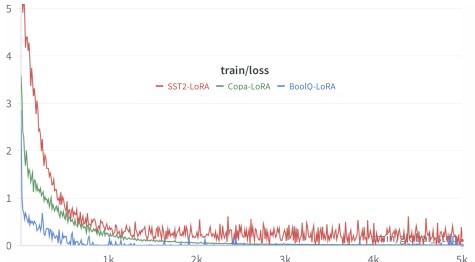

Figure 2: Training loss for the lower-level objective of the bilevel framework with Lora as the PEFT model.

and 1000 examples for testing. We use the Adam optimizer [20] and report test accuracy or F1-score.

We compare our method against several baselines, including MeZO for Full Model Fine-tuning, MeZO for PEFT, and First-order PEFT. We fix the total memory budget of each step across bilevel-ZOFO and the baselines. We train zeroth-order methods for 10,000 steps, and first-order methods

| Trainer | Mode | BoolQ | CB | Copa | ReCoRD | RTE | SST2 | WIC | WinoGrande | WSC | Average |
|---|---|---|---|---|---|---|---|---|---|---|---|
| MeZO | ft | 0.6927 | 0.7767 | 0.7000 | 0.6980 | 0.6587 | 0.8214 | 0.5543 | 0.5480 | 0.5054 | 0.6617 |
|  | lora | 0.6860 | 0.7607 | 0.7200 | 0.7083 | 0.6755 | 0.8501 | 0.5549 | 0.5607 | 0.5570 | 0.6748 |
|  | prefix | 0.6573 | 0.7945 | 0.7033 | 0.7047 | 0.6972 | 0.8218 | 0.5622 | 0.5370 | 0.5105 | 0.6654 |
|  | prompt | 0.6260 | 0.5821 | 0.7067 | 0.7070 | 0.5415 | 0.7463 | 0.5574 | 0.5556 | 0.4654 | 0.6098 |
|  | average | 0.6655 | 0.7285 | 0.7075 | 0.7045 | 0.6432 | 0.8099 | 0.5572 | 0.5503 | 0.5096 | 0.6529 |
| FO | lora | 0.7456 | 0.8512 | 0.7500 | 0.7206 | 0.7292 | 0.9258 | 0.6463 | 0.5806 | 0.6474 | 0.7330 |
|  | prefix | 0.7300 | 0.8571 | 0.7167 | 0.7093 | 0.7136 | 0.8133 | 0.5387 | 0.5787 | 0.5705 | 0.6920 |
|  | prompt | 0.7150 | 0.7142 | 0.7466 | 0.7163 | 0.6936 | 0.8016 | 0.5386 | 0.5980 | 0.5062 | 0.6700 |
|  | average | 0.7302 | 0.8075 | 0.7378 | 0.7154 | 0.7121 | 0.8470 | 0.5745 | **0.5857** | 0.5747 | 0.6977 |
| Ours | lora | 0.7433 | 0.9167 | 0.7400 | 0.7183 | 0.7401 | 0.9331 | 0.6447 | 0.5903 | 0.6428 | 0.7410 |
|  | prefix | 0.7340 | 0.8690 | 0.7267 | 0.7140 | 0.7304 | 0.8550 | 0.6317 | 0.5710 | 0.5810 | 0.7125 |
|  | prompt | 0.7367 | 0.7679 | 0.7633 | 0.7257 | 0.6867 | 0.8335 | 0.6267 | 0.5900 | 0.5133 | 0.6938 |
|  | average | **0.7380** | **0.8512** | **0.7433** | **0.7193** | **0.7191** | **0.8739** | **0.6344** | 0.5838 | **0.5790** | 0.7158 |

Table 2: Single-Task Experiments on OPT-1.3B with 1000 samples. Values correspond to mean across three random seeds. FO: First-Order. FT: full-model fine-tuning. See Table 7 in the Appendix for standard deviation values.

for 5000 steps. For all experimental details, refer to the Appendix D.1. We also provide the training loss for the lower-level objective of the bilevel framework in Figure 2 to show that consistent with the guarantees provided by our theoretical analysis in Section B, Bilevel-ZOFO converges. See Appendix D.2 for more details.

### 4.1.2 Results

**Bilevel-ZOFO mitigates MeZO's sensitivity to task prompts:** We present experimental results demonstrating that Bilevel-ZOFO significantly reduces the prompt sensitivity observed in MeZO. Following the setup of Table 5 in the MeZO paper [35], we evaluate both MeZO and Bilevel-ZOFO in two scenarios: 1- where a simple task prompt is prepended to each input versus 2- where no such prompt is used. Table 1 reports results for tuning OPT-1.3B on SST-2 and COPA using LoRA as the PEFT method. Our findings show that Bilevel-ZOFO is markedly less sensitive to prompt variations than MeZO; the performance gap between prompted and unprompted settings is substantially smaller for Bilevel-ZOFO.

| Method | Task | w/ prompt (%) | w/o prompt (%) | Diff. |
|---|---|---|---|---|
| MeZO | SST-2 | 89.6 | 51.9 | -38.6 |
|  | COPA | 70.0 | 54.8 | -15.2 |
| Ours | SST-2 | 93.3 | 92.9 | **-0.4** |
|  | COPA | 76.7 | 73.6 | **-3.1** |

Table 1: Prompt sensitivity comparison for MeZO and Bilevel-ZOFO. Bilevel-ZOFO effectively mitigates the extreme sensitivity of MeZO to adding task prompts to inputs.

Table 2 presents the test metrics when applying bilevel-ZOFO and baselines to fine-tune OPT-1.3B [62] on a downstream task. Table 3 demonstrates the results for Llama2-7b [52]. We can make the following observations:

**Bilevel-ZOFO offers a better training speed - accuracy tradeoff than MeZO** Bilevel-ZOFO outperforms MeZO, even when trained for half the number of iterations across almost all tasks, thus offering a better training duration-performance trade-off than MeZO (Also see Figure 1) .

**Bilevel-ZOFO outperforms FO PEFT on most tasks and on average:** From Table 2 and Table 3, we see that bilevel-ZOFO outperforms the corresponding FO-PEFT methods **across most instances** and **on average**, comparing each FO PEFT setting with the corresponding bilevel-ZOFO setting. This is while using the same level of memory as FO PEFT.

**Bilevel-ZOFO scales effectively to larger LLMs:** Figure 1 and Table 3 shows that bilevel-ZOFO's advantages are not confined to smaller models like OPT-1.3b, but also extend to larger LLMs.

### 4.1.3 Memory Profiling and Wall Clock Time Analysis

Figure 3 demonstrates the memory profiling of Bilevel-ZOFO, MeZO and First-order prefix tuning on four different tasks. Memory consumption of MeZO and first-order PEFT meth-

| Trainer | Mode | BoolQ | ReCoRD | SQuAD | SST2 | Average |
|---|---|---|---|---|---|---|
| MeZO | ft | 0.7915 | 0.7890 | 0.7737 | 0.8646 | 0.8047 |
|  | lora | 0.8020 | 0.7970 | 0.7412 | 0.8529 | 0.7983 |
|  | prefix | 0.7830 | 0.7905 | 0.7093 | 0.8364 | 0.7798 |
|  | prompt | 0.7787 | 0.7935 | 0.7014 | 0.8246 | 0.7746 |
|  | average | 0.7888 | 0.7925 | 0.7489 | 0.8397 | 0.7825 |
| FO | lora | 0.8420 | 0.7920 | 0.8197 | 0.9557 | 0.8524 |
|  | prefix | 0.7783 | 0.8013 | 0.7946 | 0.9243 | 0.8246 |
|  | prompt | 0.8083 | 0.8023 | 0.7805 | 0.9284 | 0.8299 |
|  | average | 0.8095 | 0.7985 | 0.7983 | 0.9361 | 0.8356 |

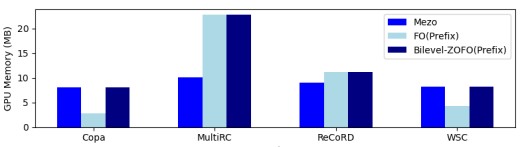

Figure 3: Memory consumption of baselines and Bilevel-ZOFO for OPT1.3B (batch size 8, A6000ada 48GB). Bilevel-ZOFO demonstrates memory usage comparable to both baselines.

| Task | MeZO | FO | Bilevel-ZOFO |
|------|------|------|--------------|
| Copa | 0.299 | 0.127 | 0.135 |
| MultiRC | 0.622 | 0.474 | 0.502 |
| WSC | 0.278 | 0.120 | 0.164 |

Table 4: Wallclock time per step of baselines and Bilevel-ZOFO when fine-tuning OPT1.3B. Values are averaged over 3 runs using a batch size of 8 on a single A6000ada 48GB GPU.

ods varies across tasks, with one occasionally surpassing the other. Each lower-level update in our method matches that of the corresponding PEFT method. Similarly, each upper-level update requires the greater memory usage between MeZO and PEFT under comparable settings. As a result, the total memory requirement of our method corresponds to the maximum memory usage of the PEFT and MeZO experiments. Nonetheless, as demonstrated in Table 2 and 3 and Figure 1, our method outperforms both PEFT and MeZO on most cases and on average.

We also present a wall-clock time analysis of bilevel-ZOFO compared to the baseline. As shown in Table 4, similar to MeZO [35], we observe that zeroth-order steps exhibit higher latency compared to first-order steps. The results indicate that our bilevel-ZOFO achieves comparable delays to the FO-PEFT method while significantly reducing step duration compared to MeZO. Moreover, as highlighted in Table 2, bilevel-ZOFO outperforms both methods on average.

## 4.2 Ablations

### 4.2.1 Effect of Hyper-parameters

We perform an ablation study by varying the regularization parameter $\lambda$ (as defined in Equation (5)) and the number of lower-level training steps between each pair of upper-level updates. Figure 4 shows the results. From Figure 4a, the effect $\lambda$ appears to be non-linear, indicating the need to find an optimal balance. Nonetheless, a moderate value like 10 or 100 seems to work reasonably well on all tasks. As anticipated, Figure 4b demonstrates that performance generally degrades when the total number of upper-level updates is reduced, suggesting there is a trade-off between latency and performance. While more upper-level updates improve results, they also extend the overall training time. We also analyze different data splits for lower and upper level training. The 1:2 split generally performs well, though effectiveness varies by task. Using a separate upper-level dataset, rather than sharing data across both levels, allows our method to adapt more quickly to new tasks in meta-learning.

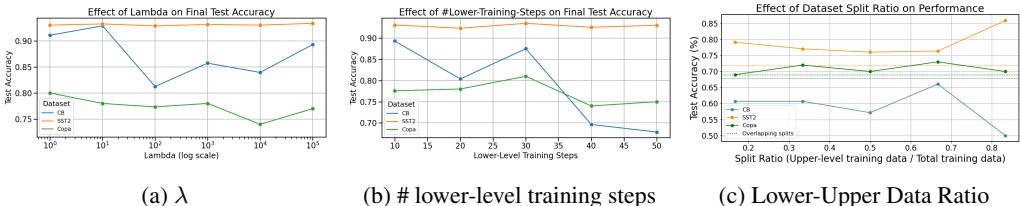

(a) $\lambda$      (b) # lower-level training steps      (c) Lower-Upper Data Ratio

Figure 4: Ablation over $\lambda$ in (5), the number of lower-level training steps before each upper-level update, and the ratio of lower/upper data.

#### 4.2.2 Effect of Design Choice

Full-model tuning is only practical via zeroth-order (ZO) optimization due to the high cost of first-order (FO) methods in large models, a core assumption of this work. To address MeZO's sensitivity and slow convergence while leveraging the strengths of FO PEFT, we propose a hybrid bilevel approach that applies FO to PEFT parameters and ZO to the base model. This section evaluates the benefits of using exact gradients for PEFT. Table 5 compares Bilevel-ZOFO with Bilevel-ZOZO, MeZO, and FO PEFT across four benchmarks, demonstrating the gains from both FO usage and the bilevel structure.

To ensure that performance gains are not simply due to tuning more parameters, we compare against a Two-Stage Pipeline baseline that is identical to Bilevel-ZOFO in parameter count, memory, and runtime. It applies FO prompt tuning for the same number of steps as Bilevel-ZOFO's lower level, followed by ZO tuning for the same number of upper-level steps. As shown in Table 5, only the bilevel formulation yields significant improvements, reinforcing the importance of the optimization structure. See Appendix D.2.1 for a detailed discussion.

| Method | BoolQ | CB | COPA | SST2 |
|---|---|---|---|---|
| MeZO | 0.6927 | 0.7767 | 0.7000 | 0.8214 |
| FO PEFT | 0.7150 | 0.7142 | 0.7466 | 0.8016 |
| Bilevel-ZOZO | 0.6280 | 0.6092 | 0.7146 | 0.7633 |
| Two-Stage Pipeline | 0.7060 | 0.6786 | 0.7433 | 0.8016 |
| Jointly Optimized | 0.7209 | 0.7500 | 0.7466 | 0.8148 |
| Bilevel-ZOFO | **0.7367** | **0.7679** | **0.7633** | **0.8335** |

Table 5: Ablation studies of the effect of different design choices of bilevel-ZOFO as well as gains from the bilevel structure itself.

We also compare with a baseline which jointly optimizes the base model parameters and PEFT parameters together with the same objective function Eq. 1 (Jointly Optimized row in Table 5). While this baseline could slightly outperform MeZO and FO PEFT, it still falls short of our bilevel method, reinforcing our approach's benefits, although it is more resource consuming and has strictly longer steps due to ZO gradient estimation at every iteration.

### 4.3 Adopting Bilevel ZOFO to Meta-learning

Following Min et al. [36], we evaluate the performance of bilevel-ZOFO as a fast and efficient meta-learning algorithm. We perform experiments using four of the distinct meta-learning settings: classification-to-classification, non-classification-to-classification, QA-to-QA, and non-QA-to-QA. For instance, in non-classification-to-classification setting, we train on a number of non-classification subtasks and test on a number of distinct classification subtasks. Each of these *meta-learning tasks* includes a set of training sub-tasks and a different set of test sub-tasks. The sub-tasks are sourced from CROSSFIT [57] and UNIFIEDQA [19], comprising a total of 142 unique sub-tasks. These sub-tasks cover a variety of problems, including text classification and question answering all in English. We use GPT2-Large [42] as the base model for these experiments.

We compare our method against several baseline approaches:
- **MetaICL** [36]: A method for meta-learning with in-context learning. MetaICL tunes all the parameters of the base model using the first-order method. In both training and testing, the model is given $k$ demonstration examples, $(a_1, b_1), \ldots, (a_k, b_k)$, where $b_i$ represents either classification labels or possible answers in question-answering tasks, along with one test example $(a, b)$. The input is formed by concatenating the demonstration examples $a_1, b_1, \ldots, a_k, b_k, a$. The model then computes the conditional probability of each label, and the label with the highest probability is selected as the prediction.
- **Zero-shot**: This method uses the pretrained language model (LM) without any tuning, performing zero-shot inference without any demonstration examples.
- **In-context Learning (ICL)**: This method uses the pretrained LM with in-context learning by conditioning on a concatenation of $k$ demonstration examples and 1 actual test sample similar to MetaICL.

We sample 768 examples from each training sub-task. We train MetaICL in their original setting for 30,000 steps. To train our method, we split the training dataset of each sub-task to two subsets, 256 samples as the development dataset for upper-level updates and 512 samples for lower-level training. For each outer iteration of our method, we randomly sample a subset of 5 training tasks. We perform 10 lower-level updates between each pair of upper-level updates. To keep bilevel-ZOFO as lightweight as possible, unlike MetaICL, we DO NOT include demonstration examples in the

| Method | Training | | | Inference | class | non_class | qa | non_qa |
|---|---|---|---|---|---|---|---|---|
| | FLOPS | Peak Mem(bs=1) | Step Duration | Tokens / Sample | $\rightarrow$ class | $\rightarrow$ class | $\rightarrow$ qa | $\rightarrow$ qa |
| Zero-shot | 0 | - | - | X | 34.2 | 34.2 | 40.2 | 40.2 |
| Few-shot | 0 | - | - | $\sim$ 5X | 34.9 (1.4) | 34.9 (1.4) | 40.5 (0.3) | 40.5 (0.4) |
| MetaICL | $1.1354 \times 10^{18}$ | 32GB | 0.48s | $\sim$ 5X | 46.4 (1.1) | 37.7 (1.7) | **45.5 (0.3)** | 40.2 (0.6) |
| Ours (Zero-shot) | $1.2485 \times 10^{18}$ | 12GB | 0.18s | X | 34.5 | 34.3 | 41.8 | 40.4 |
| Ours(Tuned) | $1.2493 \times 10^{18}$ | 12GB | 0.19s | X | **47.1** | **42.4** | 43.5 (1.3) | **41.9** |

Table 6: Multi-task Meta learning results using GPT2-Large as the base model. Values correspond to the mean and standard deviation over 5 test seeds which include different demonstration samples for each test task. class: Classification, qa: Question Answering

inputs. Since bilevel-ZOFO uses significantly less memory and has much faster updates compared to MetaICL, theoretically we are able to train it for many more iterations within the same total training duration as MetaICL. However, due to resource constraints, we only train bilevel-ZOFO for 50,000 iterations. Similar to [35], we did not observe a plateau in performance for bilevel-ZOFO, indicating that further training can yield additional improvements.

For both ICL and MetaICL, during the testing phase the model is given $k = 4$ demonstration examples for each test data point. We don't use demonstration examples in test samples for bilevel-ZOFO evaluation. We evaluate the zero-shot capabilities of our method as well as the performance of the final model LoRA-tuned for 10 additional iterations on 4 demonstration samples from each class of each test sub-task. Similar to [36], we report **Macro-averaged F1** as the evaluation metric. See Appendix D.3 for all training details.

Table 6 presents the meta-learning results. We observe that in zero-shot setting, bilevel-ZOFO (ours(zeroshot)) outperforms zero-shot on all tasks. Note that although ICL and MetaICL perform better than ours (zero-shot) 1)MetaICL fine-tunes the entire base model using first-order methods, which incurs a significantly higher computational cost. 2)both ICL and MetaICL with $k = 4$ demonstration examples take 4 times more time to do inference than our method with no demonstration examples. Nonetheless, after a lightweight 10-iteration LoRA fine-tuning phase, bilevel-ZOFO(ours(tuned)) surpasses ICL and MetaICL on nearly every hyper-task, highlighting its strong potential as a meta-learning algorithm.

# 5 Conclusions

In this work, we introduced a novel bilevel optimization framework designed to mitigate the downsides of PEFT and zeroth-order full model fine-tuning. We propose a new method that is more efficient than existing bilevel methods and thus more suitable for tuning full pre-trained large language models. Bilevel-ZOFO preserves PEFT and ZO-like peak memory, reaches target accuracy in fewer iterations than ZO (yielding 2–4× faster time-to-target despite multi-forward ZO steps), and matches or surpasses FO-PEFT at similar per-step cost—offering a practical accuracy–efficiency trade-off for resource-constrained fine-tuning. Theoretically, we provide convergence guarantees for this new method. Empirically, we show that this method outperforms both zeroth-order and FO PEFT methods in single task settings. Additionally, we show this method is effective and efficient when adapted to do multi-task learning. With competitive and even better performance compared to existing meta-training methods, our method offers a significantly cheaper training process.

# 6 Acknowledgments

This work was made possible by NSF IIS 2347592, 2348169, DBI 2405416, CCF 2348306, CNS 2347617.

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

## Limitations and Broader Impact

This paper introduces an optimization framework that enhances the efficiency of fine-tuning large language models. By reducing computational costs and memory requirements, the approach promotes environmental sustainability and broadens access to advanced AI tools, promoting accessibility in AI development. While our framework is designed for scalability, we have not tested very large LLMs due to resource constraints. However, our experiments sufficiently validate the research idea. Future work includes exploring masked ZO tuning for efficiency and applying our approach to style mixing in image generation models.

## A    Related Work

### A.1    Zeroth order in fine tuning LLMs

MeZO [35] is the first work to use Zeroth-Order (ZO) methods to finetune LLMs for downstream tasks. They demonstrate that their method is compatible with both full-parameter tuning and parameter-efficient tuning techniques, such as LoRA and prefix tuning, while being significantly more computationally efficient. Zhang et al. [63] provide a benchmark for ZO optimization in the context of LLM fine-tuning, comparing different ZO optimizers and applying the method to various models. Gautam et al. [10] introduce variance reduction techniques into ZO methods for fine-tuning, improving both stability and convergence. In addition, ZO methods are applied in federated fine-tuning by Qin et al. [41] and Ling et al. [25]. Deng et al. [5] implement ZO optimization for softmax units in LLMs. Guo et al. [14] and Liu et al. [31] explore fine-tuning a minimal subset of LLM parameters using ZO methods by sparsifying gradient approximation or the perturbations used in gradient estimation. Tang et al. [51] investigate the privacy of ZO optimization methods.

In contrast to previous approaches, we propose a bilevel training algorithm that effectively combines the strengths of both First-Order (FO) Parameter-Efficient Fine-Tuning (PEFT) and ZO full-model fine-tuning. Our experiments demonstrate that the bilevel structure, when paired with the most suitable PEFT technique, outperforms both ZO full-model fine-tuning and FO PEFT methods individually.

### A.2    Fine-tuning LLMs for Multitask and Few-Shot Learning

Multi-task learning (MTL) enables a model to handle multiple tasks simultaneously, fostering knowledge transfer between tasks and improving overall efficiency [36, 56]. Typical meta-tuning approaches employ First-Order methods to train autoregressive LLMs on a multitask dataset for various tasks [65, 36, 13, 28, 27]. Zhong et al. [65] apply meta-training to tasks such as hate speech detection, question categorization, topic classification, and sentiment classification. Guo et al. [13] adopt the method from Min et al. [36] for generating stylistic text. While Min et al. [36] focus on enhancing the in-context learning ability of the meta-trained model for multitask learning, Zhong et al. [65] focus on improving zero-shot performance. This approach is particularly valuable in low-resource settings, where collecting large labeled datasets can be costly, as is often the case with medical data. In such environments, few-shot learning—where a model is fine-tuned on a high-resource dataset to quickly adapt to new tasks with minimal data—becomes essential [57]. To address the challenges of multi-task and few-shot learning in natural language processing, several meta fine-tuning methods have been proposed [17, 64, 57, 1]. However, traditional meta fine-tuning approaches, such as MetaICL [36], still require full-model first-order gradient calculations, which become computationally expensive with large language models (LLMs) containing billions of parameters. During training, Min et al. [36] sample a task from the dataset for each iteration to perform in-context learning. In contrast to Zhong et al. [65] and Min et al. [36], our approach uses a bilevel structure: the full LLM is fine-tuned at the upper level, while parameter-efficient fine-tuning (PEFT) models are tuned at the lower level. At test time, we freeze the meta-tuned base model and fine-tune only the PEFT model using a few-shot setup, which is both more cost-effective and efficient. Crucially, Min et al. [36] fine tune the full model with first order methods, while we employ a ZO method in meta-tuning the base model at the upper level. Our approach allows us to bypass the need for backpropagation in the meta-model, significantly reducing computational costs.

## A.3 First-order Methods for Bilevel Optimization

Solving a bilevel optimization problem is challenging because the function value in the upper-level objective depends on the optimizer of the lower-level problem. This makes it difficult to compute the gradient of the upper-level objective, also known as the hypergradient. Classical methods require calculating Hessian-vector multiplications to approximate the hypergradient [8, 9, 7, 23, 43, 12, 2, 32]. However, when fine-tuning large language models, this process becomes extremely expensive due to the high computational and memory demands.

Recently, new frameworks for bilevel optimization have been introduced [33, 46, 29, 21, 26, 18, 34]. These methods bypass the need for second-order information by reformulating the bilevel problem as a constrained optimization problem. The constraint is penalized, allowing the problem to be tackled as a minimax problem using only first-order information. These methods significantly reduce computational costs by eliminating the need for second-order information. Nevertheless, when fine tuning LLMs, back propagation for calculating the gradient of an LLM is still too expensive.

Liu et al. [29] and Lu and Mei [33] explore the convergence of their proposed methods to the original bilevel problem, while other approaches only demonstrate convergence to the penalized problem. In this paper, we adapt the method from Lu and Mei [33] to approximate part of the upper-level parameters using a ZO approximation in order to address the challenge posed by the large number of training parameters in large language models. We also provide convergence guarantees for this adapted zeroth-order-first-order method.

# B Theoretical guarantees of Bilevel ZOFO

In this section we give convergence guarantee for Bilevel ZOFO. Suppose $(\boldsymbol{\theta}, \mathbf{p}) \in \mathbb{R}^{d+d'}$ and $\mathbf{s} \in \mathbb{R}^{d'}$. The following assumptions are made throughout this section.

**Assumption B.1.** We make the following assumptions:
- $G(\boldsymbol{\theta}, \mathbf{p}, \cdot)$ can be potentially nonconvex and $G(\cdot, \cdot, \mathbf{s})$ is $\tau-$ strongly concave; $F(\boldsymbol{\theta}, \mathbf{p})$ is twice continuously differentiable in $\boldsymbol{\theta}, \mathbf{p}$.
- $G$ is $\ell$-Lipschitz smooth in $\mathbb{R}^{d+2d'}$, i.e. $\forall (\boldsymbol{\theta}_1, \mathbf{p}_1, \mathbf{s}_1), (\boldsymbol{\theta}_2, \mathbf{p}_2, \mathbf{s}_2) \in \mathbb{R}^{d+2d'}$,

$$\|\nabla G(\boldsymbol{\theta}_1, \mathbf{p}_1, \mathbf{s}_1) - \nabla G(\boldsymbol{\theta}_2, \mathbf{p}_2, \mathbf{s}_2)\| \leq$$
$$\ell \|(\boldsymbol{\theta}_1, \mathbf{p}_1, \mathbf{s}_1) - (\boldsymbol{\theta}_2, \mathbf{p}_2, \mathbf{s}_2)\|.$$

  We define $\kappa := \ell/\tau$ as the problem condition number.

- $\forall (\boldsymbol{\theta}, \mathbf{p}, \mathbf{s}) \in \mathbb{R}^{d+2d'}$, sample estimates satisfy

$$\mathbb{E}[G(\boldsymbol{\theta}, \mathbf{p}, \mathbf{s}; \xi)] = G(\boldsymbol{\theta}, \mathbf{p}, \mathbf{s}),$$
$$\mathbb{E}[\nabla G(\boldsymbol{\theta}, \mathbf{p}, \mathbf{s}; \xi)] = \nabla G(\boldsymbol{\theta}, \mathbf{p}, \mathbf{s}),$$
$$\mathbb{E}\|\nabla G(\boldsymbol{\theta}, \mathbf{p}, \mathbf{s}; \xi) - \nabla G(\boldsymbol{\theta}, \mathbf{p}, \mathbf{s})\|^2 \leq \frac{\sigma^2}{B}$$

  for sample $\xi$ with size $|\xi| = B$ and constant $\sigma > 0$.

- $\max_{\mathbf{s}} G(\boldsymbol{\theta}, \mathbf{p}, \mathbf{s})$ is lower bounded.

We first discuss the relationship between the optimality condition (4) and (2). We start with defining the $\epsilon$-stationary points of (4) and (2) for general bilevel and minimax problems. In the following definitions, the expectation is taken over the randomness in the algorithm that $(\mathbf{x}, \mathbf{y})$ is generated.

**Definition B.2.** Given a bilevel optimization problem

$$f^* = \min_{\mathbf{x}} f(\mathbf{x}, \mathbf{y}^*(\mathbf{x})), \mathbf{y}^*(\mathbf{x}) \in \arg\min_{\mathbf{z}} g(\mathbf{x}, \mathbf{z})$$

and any $\epsilon > 0$, a point $(\mathbf{x}_\epsilon, \mathbf{y}_\epsilon)$ is called an $\epsilon$-stationary point if

$$\mathbb{E}[\|\nabla f(\mathbf{x}_\epsilon, \mathbf{y}^*(\mathbf{x}_\epsilon))\|] \leq O(\epsilon), f(\mathbf{x}_\epsilon, \mathbf{y}_\epsilon) - \min_{\mathbf{z}} f(\mathbf{x}_\epsilon, \mathbf{z}) \leq \epsilon.$$

**Definition B.3.** Given a minimax problem

$$f^* = \min_{\mathbf{x}} \max_{\mathbf{y}} f(\mathbf{x}, \mathbf{y})$$

and any $\epsilon > 0$, a point $(\mathbf{x}_\epsilon, \mathbf{y}_\epsilon)$ is called an $\epsilon$-stationary point if

$$\mathbb{E}[\|\nabla_\mathbf{x} f(\mathbf{x}_\epsilon, \mathbf{y}_\epsilon)\|^2] \leq \epsilon^2, \ \mathbb{E}[\|\nabla_\mathbf{y} f(\mathbf{x}_\epsilon, \mathbf{y}_\epsilon)\|^2] \leq \epsilon^2.$$

**Lemma B.4.** *If assumption B.1 holds and $\lambda = 1/\epsilon$, assume that $\nabla^2 F(\boldsymbol{\theta}, \cdot)$ is Lipschitz continuous and $(\boldsymbol{\theta}, \mathbf{p}, \mathbf{s})$ is an $\epsilon$-stationary point of (4), then $(\boldsymbol{\theta}, \mathbf{s})$ is an $\epsilon$-stationary point of (2).*

The following is the low effective rank assumption from [35]. This assumption avoids dimension $d$ in the total complexity. Following [35], we assume here that $\mathbf{z}^k$ in (7) is sampled from shpere in $\mathbb{R}^d$ with radius $\sqrt{d}$ for ease of illustration.

**Assumption B.5.** For any $(\boldsymbol{\theta}, \mathbf{p}, \mathbf{s}) \in \mathbb{R}^{d+2d'}$, there exists a matrix $H(\boldsymbol{\theta}, \mathbf{p}, \mathbf{s})$ such that $\nabla^2 G(\boldsymbol{\theta}, \mathbf{p}, \mathbf{s}) \preceq H(\boldsymbol{\theta}, \mathbf{p}, \mathbf{s}) \preceq \ell \cdot I_d$ and $tr(H(\boldsymbol{\theta}, \mathbf{p}, \mathbf{s})) \leq r \cdot \|H(\boldsymbol{\theta}, \mathbf{p}, \mathbf{s})\|$.

**Theorem B.6.** *If Assumptions B.1 and B.5 hold, by setting*

$$\eta = \frac{1}{2\ell}, \zeta = \frac{1}{2\ell r}, \lambda = \frac{1}{\epsilon}, B = O(\sigma^2 \epsilon^{-2}),$$
$$\alpha = O(\epsilon \kappa^{-1}(d + d')^{-1.5}), T = O\left(\kappa \log(\kappa \epsilon^{-1})\right),$$
$$K = O(\kappa r \epsilon^{-2})$$

*there exists an iteration in Algorithm 2 that returns an $\epsilon$-stationary point $(\boldsymbol{\theta}, \mathbf{p}, \mathbf{s})$ for (5) and it satisfies*

$$\mathbb{E}[\|\nabla F(\boldsymbol{\theta}, \mathbf{p}^*(\boldsymbol{\theta}); \mathcal{D}_f)\|] \leq O(\epsilon),$$
$$F(\boldsymbol{\theta}, \mathbf{s}; \mathcal{D}_\mathbf{p}) - \min_\mathbf{p} F(\boldsymbol{\theta}, \mathbf{p}; \mathcal{D}_\mathbf{p}) \leq \epsilon.$$

*Remark* B.7. The total number of ZO gradient calculations is

$$TKB_1 + KB_2 = O(\sigma^2 \kappa^2 r \epsilon^{-4} \log(\kappa \epsilon^{-1})).$$

This result matches the complexity in previous ZO minimax algorithm in [55] but solves our bilevel optimization problem (2) and does not depend on the dimensionality $d$ thanks to the efficient rank assumption B.5, providing efficiency guarantee for our algorithm.

## C  Method

### C.1  Proofs

In the proofs we use the simplified notations $\mathbf{x} := (\boldsymbol{\theta}, \mathbf{p})$, $\mathbf{y} := \mathbf{s}$, $f(\mathbf{x}, \mathbf{y}) := G(\boldsymbol{\theta}, \mathbf{p}, \mathbf{s})$, $\mathbf{y}^*(\mathbf{x}) := \arg\max_\mathbf{y} f(\mathbf{x}, \mathbf{y})$ and $g(\mathbf{x}) := f(\mathbf{x}, \mathbf{y}^*(\mathbf{x}))$.

#### C.1.1  proof of lemma B.4

First we introduce some lemmas from previous literature.

**Lemma C.1.** *(Lemma 1.2.3, Theorem 2.1.8 and Theorem 2.1.10 in [37])*

- *Suppose a function $h$ is $L_h$-gradient-Lipschitz and has a unique maximizer $\mathbf{x}^*$. Then, for any $\mathbf{x}$, we have:*

$$\frac{1}{2L_h}\|\nabla h(\mathbf{x})\|_2^2 \leq h(\mathbf{x}^*) - h(\mathbf{x}) \leq \frac{L_h}{2}\|\mathbf{x} - \mathbf{x}^*\|_2^2. \tag{15}$$

- *Suppose a function $h$ is $\tau_h$-strongly concave and has a unique maximizer $\mathbf{x}^*$. Then, for any $\mathbf{x}$, we have:*

$$\frac{\tau_h}{2}\|\mathbf{x} - \mathbf{x}^*\|_2^2 \leq h(\mathbf{x}^*) - h(\mathbf{x}) \leq \frac{1}{2\tau_h}\|\nabla h(\mathbf{x})\|_2^2. \tag{16}$$

From lemma C.1 and the definition of $\epsilon$-stationary point (in definition B.3) we can get the following lemma.

**Lemma C.2.** *Suppose assumption B.1 holds and $(\mathbf{x}_\epsilon, \mathbf{y}_\epsilon)$ is an $\epsilon$-stationary point of $\min_\mathbf{x} \max_\mathbf{y} f(\mathbf{x}, \mathbf{y})$, let $(\boldsymbol{\theta}_\epsilon, \mathbf{p}_\epsilon) = \mathbf{x}_\epsilon$ we have*

$$F(\boldsymbol{\theta}_\epsilon, \mathbf{s}_\epsilon) - \min_\mathbf{s} F(\boldsymbol{\theta}_\epsilon, \mathbf{s}) \leq O(\frac{\epsilon^2}{\lambda^2}).$$

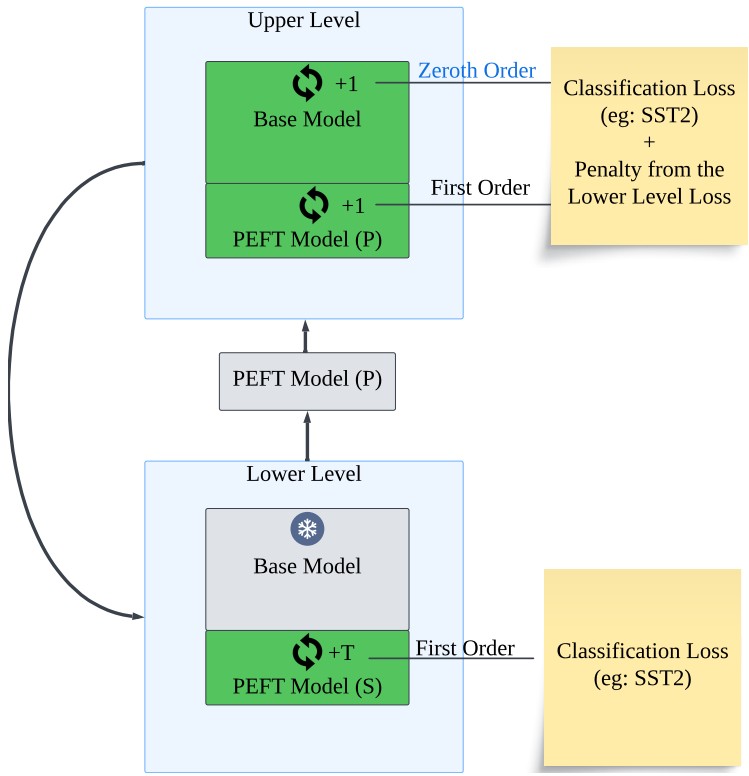

Figure 5: Bilevel ZOFO optimizes LLM fine-tuning by solving a bilevel problem using a penalty-based minimax approach, combining zeroth-order gradient estimation for LLM updates and first-order methods for PEFT parameters.

*Proof.*

$$F(\boldsymbol{\theta}_\epsilon, \mathbf{s}_\epsilon) - \min_{\mathbf{s}} F(\boldsymbol{\theta}_\epsilon, \mathbf{s}) \leq \frac{1}{\tau}\|\nabla_{\mathbf{s}} F(\boldsymbol{\theta}_\epsilon, \mathbf{s}_\epsilon)\|^2 = \frac{1}{\lambda^2\tau}\|\nabla_{\mathbf{y}} f(\mathbf{x}_\epsilon, \mathbf{y}_\epsilon)\|^2 \leq O(\frac{\epsilon^2}{\lambda^2}),$$

here the first inequality is from Lemma C.1 applied to $-F$ and the second inequality from definition B.3. $\qquad\square$

The following is a rephrase of theorem 2 in [33].

*Proof.* (proof of lemma B.4) By Lemma C.2 and the value of $\lambda$ we have

$$F(\boldsymbol{\theta}_\epsilon, \mathbf{s}_\epsilon) - \min_{\mathbf{s}} F(\boldsymbol{\theta}_\epsilon, \mathbf{s}) \leq O(\epsilon^4).$$

Therefore, by Theorem 2 in [33] we have $\mathbb{E}[\|\nabla F(\boldsymbol{\theta}, \mathbf{p}^*(\boldsymbol{\theta}))\|] \leq O(\epsilon)$ and Lemma B.4 is proven. $\quad\square$

### C.1.2 proof of theorem B.6

Based on Lemma B.4, it suffices to prove that the algorithm 2 outputs an $\epsilon$-stationary point of $\min_{\mathbf{x}} \max_{\mathbf{y}} f(\mathbf{x}, \mathbf{y})$. In this section we will prove this conclusion.

First we introduce the smoothed function of $f$, which will be useful in the proof.

**Lemma C.3.** *(Lemma C.2 in [60]) Let $\mathbf{u}$ be uniformly sampled from the Euclidean sphere $\sqrt{d}\mathbf{s}^{d-1}$ and $\mathbf{v}$ be uniformly sampled from the Euclidean ball $\sqrt{d}\mathbb{B}^d = \{\mathbf{x} \in \mathbb{R}^d \mid \|\mathbf{x}\| \leq \sqrt{d}\}$. For any function $f(\mathbf{x}) : \mathbb{R}^d \to \mathbb{R}$ and $\alpha > 0$, we define its zeroth-order gradient estimator as:*

$$\hat{\nabla} f_\alpha(\mathbf{x}) = \frac{f(\mathbf{x} + \alpha\mathbf{u}) - f(\mathbf{x} - \alpha\mathbf{u})}{2\alpha}\mathbf{u},$$

*and the smoothed function as:*

$$f_\alpha(\mathbf{x}) = \mathbb{E}_\mathbf{v}[f(\mathbf{x} + \alpha\mathbf{v})].$$

*The following properties hold:*

(i) $f_\alpha(\mathbf{x})$ *is differentiable and* $\mathbb{E}_\mathbf{u}[\hat{\nabla}f_\alpha(\mathbf{x})] = \nabla f_\alpha(\mathbf{x})$.

(ii) *If* $f(\mathbf{x})$ *is* $\ell$-smooth, then we have that:

$$\|\nabla f(\mathbf{x}) - \nabla f_\alpha(\mathbf{x})\| \le \frac{\ell}{2}\alpha d^{3/2}.$$

If we use $f(\mathbf{x}, \mathbf{y}; \xi)$ to denote a forward evaluation with random samples $\xi$ and let batch size $B = |\xi|$, then $f(\mathbf{x}, \cdot; \xi)$ is a function from $\mathbb{R}^d$ to $\mathbb{R}$ and $\ell$-smooth. The above lemma can be used on $f(\mathbf{x}, \cdot)$ and $f(\mathbf{x}, \cdot; \xi)$. We can define its smoothed function $f_\alpha(\mathbf{x}, \cdot; \xi)$ and has the properties above.

**Lemma C.4.** *If assumption B.1 holds, for* $f_\alpha$ *defined in Lemma C.3,* $\nabla_\mathbf{x} f_\alpha(\mathbf{x}, \mathbf{y})$ *is* $\ell$-continuous on $\mathbf{y}$, *i.e.*

$$\|\nabla_\mathbf{x} f_\alpha(\mathbf{x}, \mathbf{y}_1) - \nabla_\mathbf{x} f_\alpha(\mathbf{x}, \mathbf{y}_2)\| \le \ell\|\mathbf{y}_1 - \mathbf{y}_2\|,$$

*for any* $\mathbf{x} \in \mathbb{R}^d, \mathbf{y}_1, \mathbf{y}_2 \in \mathbb{R}^{d'}$.

*Proof.*

$$\begin{aligned}
&\|\nabla_\mathbf{x} f_\alpha(\mathbf{x}, \mathbf{y}_1) - \nabla_\mathbf{x} f_\alpha(\mathbf{x}, \mathbf{y}_2)\| \\
=&\|\mathbb{E}_\mathbf{v}[f(\mathbf{x} + \alpha\mathbf{v}, \mathbf{y}_1)] - \mathbb{E}_\mathbf{v}[f(\mathbf{x} + \alpha\mathbf{v}, \mathbf{y}_2)]\| \\
\le&\mathbb{E}_\mathbf{v}\|f(\mathbf{x} + \alpha\mathbf{v}, \mathbf{y}_1) - f(\mathbf{x} + \alpha\mathbf{v}, \mathbf{y}_2)\| \\
\le&\ell\|\mathbf{y}_1 - \mathbf{y}_2\|.
\end{aligned}$$

Here the first inequality is from the convexity of norm and the second inequality is from the $\ell$-smoothness of $f$. $\qquad\square$

We first give the iteration complexity of the inner loop of Algorithm 2. Using the simplified notations we can write the update step in the inner loop as $\mathbf{y}_{t+1}^k = \mathbf{y}_t^k + \eta\nabla_\mathbf{y} f(\mathbf{x}^k, \mathbf{y}_t^k; \xi_t)$. We use $B_1, B_2$ to denote the batch size for the inner loop and outer loop, respectively. But finally we will prove that they are in fact of the same order.

**Lemma C.5.** *In Algorithm 2, by setting* $\eta = 1/2\ell$, $T = O(\kappa\log(\frac{1}{\epsilon}))$ *and* $B_1 = O(\epsilon^{-2})$ *we have*

$$\mathbb{E}[\|\mathbf{y}_T^k - \mathbf{y}^*(\mathbf{x}^k)\|^2] \le \epsilon^2$$

*in outer loop* $k$.

*Proof.*

$$\begin{aligned}
&\|\mathbf{y}_{t+1}^k - \mathbf{y}^*(\mathbf{x}^k)\|^2 \\
=&\|\mathbf{y}_t^k + \eta\nabla_\mathbf{y} f(\mathbf{x}^k, \mathbf{y}_t^k; \xi_t) - \mathbf{y}^*(\mathbf{x}^k)\|^2 \\
=&\|\mathbf{y}_t^k - \mathbf{y}^*(\mathbf{x}^k)\|^2 + 2\eta\langle\nabla_\mathbf{y} f(\mathbf{x}^k, \mathbf{y}_t^k; \xi_t), \mathbf{y}_t^k - \mathbf{y}^*(\mathbf{x}^k)\rangle + \eta^2\|\nabla_\mathbf{y} f(\mathbf{x}^k, \mathbf{y}_t^k; \xi_t)\|^2.
\end{aligned}$$

Now taking expectations on both sides we have

$$\begin{aligned}
&\mathbb{E}[\|\mathbf{y}_{t+1}^k - \mathbf{y}^*(\mathbf{x}^k)\|^2] \\
\le&\mathbb{E}[\|\mathbf{y}_t^k - \mathbf{y}^*(\mathbf{x}^k)\|^2] + 2\eta\mathbb{E}[\langle\nabla_\mathbf{y} f(\mathbf{x}^k, \mathbf{y}_t^k), \mathbf{y}_t^k - \mathbf{y}^*(\mathbf{x}^k)\rangle] + \eta^2(\mathbb{E}[\|\nabla_\mathbf{y} f(\mathbf{x}^k, \mathbf{y}_t^k)\|^2] + \frac{\sigma^2}{B_1}) \\
\le&\mathbb{E}[\|\mathbf{y}_t^k - \mathbf{y}^*(\mathbf{x}^k)\|^2] - 2\eta\mathbb{E}[f(\mathbf{x}^k, \mathbf{y}^*(\mathbf{x}^k)) - f(\mathbf{x}^k, \mathbf{y}_t^k)] + 2\ell\eta^2\mathbb{E}[f(\mathbf{x}^k, \mathbf{y}^*(\mathbf{x}^k)) - f(\mathbf{x}^k, \mathbf{y}_t^k)] + \frac{\eta^2\sigma^2}{B_1} \\
=&\mathbb{E}[\|\mathbf{y}_t^k - \mathbf{y}^*(\mathbf{x}^k)\|^2] - \frac{1}{2\ell}\mathbb{E}[f(\mathbf{x}^k, \mathbf{y}^*(\mathbf{x}^k)) - f(\mathbf{x}^k, \mathbf{y}_t^k)] + \frac{\sigma^2}{4\ell^2 B_1} \\
\le&\mathbb{E}[\|\mathbf{y}_t^k - \mathbf{y}^*(\mathbf{x}^k)\|^2] - \frac{\tau}{4\ell}\mathbb{E}[\|\mathbf{y}_t^k - \mathbf{y}^*(\mathbf{x}^k)\|^2] + \frac{\sigma^2}{4\ell^2 B_1}.
\end{aligned}$$

The first inequality is from Assumption B.1, second and last inequalities from Lemma C.1 and the equation is from the value of $\eta$.

In order for $\mathbb{E}[\|\mathbf{y}_T^k - \mathbf{y}^*(\mathbf{x}^k)\|^2] \leq \epsilon^2$ we need $T = O(\kappa \log(\frac{1}{\epsilon}))$ and $B_1 = O(\epsilon^{-2})$. $\qquad\square$

The following lemma is from Theorem 1 in [35].

**Lemma C.6.** *If Assumption B.5 holds, there exists a constant $\gamma = \boldsymbol{\theta}(r)$ such that*

$$\mathbb{E}[\hat{\nabla}_{\mathbf{x}} f(\mathbf{x}^k, \mathbf{y}^{k+1}; \xi)^T H(\mathbf{x}^k, \mathbf{y}^{k+1}) \hat{\nabla}_{\mathbf{x}} f(\mathbf{x}^k, \mathbf{y}^{k+1}; \xi)] \leq \ell \gamma \mathbb{E}[\|\nabla_{\mathbf{x}} f(\mathbf{x}^k, \mathbf{y}^{k+1}; \xi)\|^2].$$

Finally, we give the proof for Theorem B.6. In this part we assume both $\boldsymbol{\theta}$ and $\mathbf{p}$ updates with zeroth order gradient for the convenience of analysis and this does not change the order of the total complexity.

*Proof.* (**proof of Theorem B.6**)

From Assumption B.5, taking expectation conditioning on $\mathbf{x}^k$ and $\mathbf{y}^{k+1}$ we have

$$
\begin{aligned}
\mathbb{E}[g(\mathbf{x}^{k+1})] \leq & g(\mathbf{x}^k) - \zeta \langle \nabla_{\mathbf{x}} g(\mathbf{x}^k), \mathbb{E}[\hat{\nabla}_{\mathbf{x}} f(\mathbf{x}^k, \mathbf{y}^{k+1}; \xi)] \rangle \\
& + \frac{\zeta^2}{2} \mathbb{E}[\hat{\nabla}_{\mathbf{x}} f(\mathbf{x}^k, \mathbf{y}^{k+1}; \xi)^T H(\mathbf{x}^k, \mathbf{y}^{k+1}) \hat{\nabla}_{\mathbf{x}} f(\mathbf{x}^k, \mathbf{y}^{k+1}; \xi)] \\
\leq & g(\mathbf{x}^k) - \zeta \langle \nabla_{\mathbf{x}} g(\mathbf{x}^k), \nabla_x f_\alpha(\mathbf{x}^k, \mathbf{y}^{k+1}) \rangle + \frac{\zeta^2}{2} \ell \gamma \mathbb{E}[\|\nabla_{\mathbf{x}} f(\mathbf{x}^k, \mathbf{y}^{k+1}; \xi)\|^2]
\end{aligned}
$$

Let us bound the inner product term:

$$
\begin{aligned}
& - \zeta \langle \nabla_{\mathbf{x}} g(\mathbf{x}^k), \nabla_x f_\alpha(\mathbf{x}^k, \mathbf{y}^{k+1}) \rangle \\
\leq & - \zeta \langle \nabla_{\mathbf{x}} f(\mathbf{x}^k, \mathbf{y}^*(\mathbf{x}^k)) - \nabla_{\mathbf{x}} f_\alpha(\mathbf{x}^k, \mathbf{y}^*(\mathbf{x}^k)) + \nabla_{\mathbf{x}} f_\alpha(\mathbf{x}^k, \mathbf{y}^*(\mathbf{x}^k)) \\
& - \nabla_x f_\alpha(\mathbf{x}^k, \mathbf{y}^{k+1}) + \nabla_x f_\alpha(\mathbf{x}^k, \mathbf{y}^{k+1}), \nabla_x f_\alpha(\mathbf{x}^k, \mathbf{y}^{k+1}) \rangle \\
\leq & \frac{1}{\ell\gamma} \|\nabla_{\mathbf{x}} f(\mathbf{x}^k, \mathbf{y}^*(\mathbf{x}^k)) - \nabla_{\mathbf{x}} f_\alpha(\mathbf{x}^k, \mathbf{y}^*(\mathbf{x}^k))\|^2 + \frac{\zeta^2 \ell\gamma}{4} \|\nabla_x f_\alpha(\mathbf{x}^k, \mathbf{y}^{k+1})\|^2 \\
& + \frac{1}{\ell\gamma} \|\nabla_{\mathbf{x}} f_\alpha(\mathbf{x}^k, \mathbf{y}^*(\mathbf{x}^k)) - \nabla_x f_\alpha(\mathbf{x}^k, \mathbf{y}^{k+1})\|^2 + \frac{\zeta^2 \ell\gamma}{4} \|\nabla_x f_\alpha(\mathbf{x}^k, \mathbf{y}^{k+1})\|^2 \\
& - \zeta \langle \nabla_x f_\alpha(\mathbf{x}^k, \mathbf{y}^{k+1}), \nabla_x f_\alpha(\mathbf{x}^k, \mathbf{y}^{k+1}) \rangle \\
\leq & \frac{\alpha^2 \ell^2 d^3}{4\ell\gamma} + \frac{\ell^2}{\ell\gamma} \|\mathbf{y}^*(\mathbf{x}^k) - \mathbf{y}^{k+1}\|^2 + \frac{\zeta^2 \ell\gamma}{2} \|\nabla_x f_\alpha(\mathbf{x}^k, \mathbf{y}^{k+1})\|^2 \\
& - \zeta \langle \nabla_x f_\alpha(\mathbf{x}^k, \mathbf{y}^{k+1}), \nabla_x f_\alpha(\mathbf{x}^k, \mathbf{y}^{k+1}) \rangle.
\end{aligned}
$$

Here the last inequality is from Lemma C.3 and Lemma C.4.

Now back to the original inequality, taking expectations over all the randomness in the algorithm we have

$$
\begin{aligned}
& \zeta(1 - \frac{\zeta\ell\gamma}{2}) \mathbb{E}[\|\nabla_x f_\alpha(\mathbf{x}^k, \mathbf{y}^{k+1})\|^2] \\
\leq & \mathbb{E}[g(\mathbf{x}^k) - g(\mathbf{x}^{k+1})] + \frac{\ell}{\gamma} \mathbb{E}[\|\mathbf{y}^*(\mathbf{x}^k) - \mathbf{y}^{k+1}\|^2] + \frac{\zeta^2\ell\gamma}{2} \mathbb{E}[\|\nabla_{\mathbf{x}} f(\mathbf{x}^k, \mathbf{y}^{k+1}; \xi)\|^2] + \frac{\alpha^2 \ell d^3}{4\gamma} \\
\leq & \mathbb{E}[g(\mathbf{x}^k) - g(\mathbf{x}^{k+1})] + \frac{\ell}{\gamma} \mathbb{E}[\|\mathbf{y}^*(\mathbf{x}^k) - \mathbf{y}^{k+1}\|^2] + \frac{\zeta^2\ell\gamma}{2} \mathbb{E}[\|\nabla_{\mathbf{x}} f(\mathbf{x}^k, \mathbf{y}^{k+1}; \xi)\|^2] + \frac{\zeta^2\ell\gamma\sigma^2}{2B_2} + \frac{\alpha^2 \ell d^3}{4\gamma},
\end{aligned}
$$

where the last inequality is from Assumption B.1.

On the other hand, from Lemma C.3, by letting $\zeta = \frac{1}{2\ell\gamma}$ we have

$$\mathbb{E}[\|\nabla_x f(\mathbf{x}^k, \mathbf{y}^{k+1})\|^2]$$

$$\leq 2\mathbb{E}[\|\nabla_x f_\alpha(\mathbf{x}^k, \mathbf{y}^{k+1})\|^2] + \frac{\alpha^2\ell^2(d+d')^3}{2}$$

$$\leq \frac{16}{3}\ell\gamma\mathbb{E}[g(\mathbf{x}^k) - g(\mathbf{x}^{k+1})] + \frac{16}{3}\ell^2\mathbb{E}[\|\mathbf{y}^*(\mathbf{x}^k) - \mathbf{y}^{k+1}\|^2]$$

$$+ \frac{2}{3}\mathbb{E}[\|\nabla_\mathbf{x} f(\mathbf{x}^k, \mathbf{y}^{k+1})\|^2] + \frac{2\sigma^2}{3B_2} + \frac{11}{6}\alpha^2\ell^2(d+d')^3$$

$$\Rightarrow \mathbb{E}[\|\nabla_x f(\mathbf{x}^k, \mathbf{y}^{k+1})\|^2] \leq 16\ell\gamma\mathbb{E}[g(\mathbf{x}^k) - g(\mathbf{x}^{k+1})] + 16\ell^2\mathbb{E}[\|\mathbf{y}^*(\mathbf{x}^k) - \mathbf{y}^{k+1}\|^2]$$

$$+ \frac{2\sigma^2}{B_2} + \frac{11}{2}\alpha^2\ell^2(d+d')^3.$$

Taking summation of $k$ from 1 to $K$ we have

$$\frac{1}{K}\sum_{k=1}^{K+1}\mathbb{E}[\|\nabla_x f(\mathbf{x}^k, \mathbf{y}^{k+1})\|^2]$$

$$\leq \frac{16\ell\gamma}{K}\mathbb{E}[g(\mathbf{x}^1) - g(\mathbf{x}^{K+1})] + \frac{16\ell^2}{K}\sum_{k=1}^{K}\mathbb{E}[\|\mathbf{y}^*(\mathbf{x}^k) - \mathbf{y}^{k+1}\|^2] + \frac{2\sigma^2}{B_2} + \frac{11}{2}\alpha^2\ell^2(d+d')^3$$

$$\leq \frac{16\ell\gamma}{K}\mathbb{E}[g(\mathbf{x}^1) - \min_\mathbf{x} g(\mathbf{x})] + \frac{16\ell^2}{K}\sum_{k=1}^{K}\mathbb{E}[\|\mathbf{y}^*(\mathbf{x}^k) - \mathbf{y}^{k+1}\|^2] + \frac{2\sigma^2}{B_2} + \frac{11}{2}\alpha^2\ell^2(d+d')^3.$$

Thus, by setting parameters as in Theorem B.6 we have $\min_k \mathbb{E}[\|\nabla_x f(\mathbf{x}^k, \mathbf{y}^{k+1})\|^2] \leq \epsilon^2$.

On the other hand, since

$$\mathbb{E}[\|\nabla_x f(\mathbf{x}^k, \mathbf{y}^{k+1})\|^2] = \mathbb{E}[\|\nabla_x f(\mathbf{x}^k, \mathbf{y}^{k+1}) - \nabla_\mathbf{y} f(\mathbf{x}^k, \mathbf{y}^*(\mathbf{x}^k))\|^2] \leq \ell^2\mathbb{E}[\|\mathbf{y}^{k+1} - \mathbf{y}^*(\mathbf{x}^k)\|^2],$$

similar to Lemma C.5 we have $\mathbb{E}[\|\nabla_x f(\mathbf{x}^k, \mathbf{y}^{k+1})\|^2] \leq \epsilon^2$ by setting $T = O(\kappa \log(\frac{\kappa}{\epsilon}))$ and $B_1 = O(\epsilon^{-2})$. $\qquad\square$

# D   Experimental Setup

## D.1   Single-Task experiments

Following MeZO [35], we evaluate our approach on a range of classification and multiple-choice tasks: BoolQ [3], CB [54], CB [54], COPA [44], ReCoRD: [61],RTE [53], SST2 [53], WiC [39], WinoGrande [45]. In this setting, training and testing are conducted on the same task.

**Hyperparameter Search** Given resource limitations, we focus on sweeping only the learning rate as the key hyperparameter. For MeZO and first-order PEFT experiments, we explore learning rates from the set $\{1e-2, 1e-3, 1e-4, 1e-5, 1e-6\}$. For Bilevel-ZOFO, we sweep both the upper-level and lower-level learning rates: $\mathrm{lr}_{\mathrm{upper}} \in \{1e-4, 1e-5, 1e-6\}$ and $\mathrm{lr}_{\mathrm{lower}} \in \{1e-2, 1e-3, 1e-4, 1e-5\}$. We perform all experiments in tables 7 and 8 using three random seeds and report the average and standard deviation. We also set $\epsilon = 1e-3$, following MeZO [35].

### D.1.1   Training

All experiments used a batch size of 8 and were conducted in bfloat16 precision on a single A6000 Ada 48GB GPU. MeZO was run for 10,000 steps, while FO and Bilevel-ZOFO methods were run for 5,000 steps. Our implementation builds upon MeZO's codebase, and memory profiling as well as latency calculations are based on their framework.

For each task, 1000 examples are randomly sampled for training, 500 for validation, and 1000 for testing. For bilevel-ZOFO, the training set is split into upper-level and lower-level subsets with a 1:2 ratio. During each lower-level update, only the PEFT parameters are optimized, while in the

upper-level step, the entire model is fine-tuned using zeroth-order gradient approximation. We set $\lambda = 10000$ and perform 10 lower-level updates between each upper-level update for all bilevel-ZOFO experiments.

All experiments use the Adam optimizer [20],including baselines and both lower-level and upper-level optimizers. No weight decay was applied, and the models were trained with a constant learning rate schedule. Batch size is set to 16 for all experiments. We load all models in bfloat16. We find the best performing model based on validation loss and report test results from that checkpoint. We report the test accuracy or F1-score based on the test dataset being imbalanced or not.

We fix the memory budget of each step across bilevel-ZOFO and the baselines. We train zeroth-order methods for 10,000 steps, and bilevel-ZOFO and first-order methods for 5000 steps. We use A6000ada 48GPUs in our experiments. We load all models in bfloat16.

## D.2  More Results for single task fine tuning

Table 7 presents the detailed test metrics when applying bilevel-ZOFO and baselines to fine-tune OPT-1.3B [62] on a downstream task.

| Trainer | Mode | BoolQ | CB | Copa | ReCoRD | RTE | SST2 | WIC | WinoGrande | WSC | Average |
|---|---|---|---|---|---|---|---|---|---|---|---|
| MeZO | ft | 0.6927 ± 0.0660 | 0.7767 ± 0.1162 | 0.7000 ± 0.0289 | 0.6980 ± 0.0053 | 0.6587 ± 0.0271 | 0.8214 ± 0.0042 | 0.5543 ± 0.0146 | 0.5480 ± 0.0108 | 0.5054 ± 0.0056 | 0.6617 ± 0.0321 |
| | lora | 0.6860 ± 0.0012 | 0.7607 ± 0.0515 | 0.7200 ± 0.0058 | 0.7083 ± 0.0049 | 0.6755 ± 0.0110 | 0.8501 ± 0.0067 | 0.5549 ± 0.0057 | 0.5607 ± 0.0050 | 0.5570 ± 0.0000 | 0.6748 ± 0.0102 |
| | prefix | 0.6573 ± 0.0379 | 0.7945 ± 0.0309 | 0.7033 ± 0.0208 | 0.7047 ± 0.0010 | 0.6972 ± 0.0055 | 0.8218 ± 0.0127 | 0.5622 ± 0.0127 | 0.5370 ± 0.0137 | 0.5105 ± 0.1313 | 0.6654 ± 0.0285 |
| | prompt | 0.6260 ± 0.0056 | 0.5821 ± 0.0179 | 0.7067 ± 0.0058 | 0.7070 ± 0.0053 | 0.5415 ± 0.0063 | 0.7463 ± 0.0218 | 0.5574 ± 0.0048 | 0.5556 ± 0.0038 | 0.4654 ± 0.0618 | 0.6098 ± 0.0159 |
| | average | 0.6655 | 0.7285 | 0.7075 | 0.7045 | 0.6432 | 0.8099 | 0.5572 | 0.5503 | 0.5096 | 0.6529 ± 0.0217 |
| FO | lora | 0.7403 ± 0.0055 | 0.8512 ± 0.0412 | 0.7500 ± 0.0058 | 0.7206 ± 0.0035 | 0.7292 ± 0.0165 | 0.9258 ± 0.0032 | 0.6463 ± 0.0276 | 0.5806 ± 0.0055 | 0.6474 ± 0.0200 | 0.7324 ± 0.0143 |
| | prefix | 0.7300 ± 0.0035 | 0.8571 ± 0.0644 | 0.7167 ± 0.0115 | 0.7093 ± 0.0032 | 0.7136 ± 0.0110 | 0.8133 ± 0.0050 | 0.5387 ± 0.0050 | 0.5980 ± 0.0029 | 0.5705 ± 0.0294 | 0.6941 ± 0.0141 |
| | prompt | 0.7150 ± 0.0156 | 0.7142 ± 0.0714 | 0.7466 ± 0.0115 | 0.7163 ± 0.0063 | 0.6936 ± 0.0185 | 0.8016 ± 0.0779 | 0.5386 ± 0.0197 | 0.5980 ± 0.0090 | 0.5062 ± 0.0434 | 0.6700 ± 0.0306 |
| | average | 0.7284 | 0.8075 | 0.7378 | 0.7154 | 0.7121 | 0.8470 | 0.5745 | 0.5922 | 0.5747 | 0.6982 ± 0.0197 |
| Ours | lora | 0.7433 ± 0.0191 | 0.9167 ± 0.0103 | 0.7400 ± 0.0200 | 0.7183 ± 0.0031 | 0.7401 ± 0.0108 | 0.9331 ± 0.0020 | 0.6447 ± 0.0218 | 0.5903 ± 0.0058 | 0.6428 ± 0.0855 | 0.7410 ± 0.0209 |
| | prefix | 0.7340 ± 0.0095 | 0.8690 ± 0.0206 | 0.7267 ± 0.0153 | 0.7140 ± 0.0044 | 0.7304 ± 0.0091 | 0.8550 ± 0.0178 | 0.6317 ± 0.0282 | 0.5710 ± 0.0130 | 0.5810 ± 0.0338 | 0.7125 ± 0.0179 |
| | prompt | 0.7367 ± 0.0850 | 0.7679 ± 0.0644 | 0.7633 ± 0.0058 | 0.7257 ± 0.0153 | 0.6867 ± 0.0208 | 0.8335 ± 0.0779 | 0.6267 ± 0.0462 | 0.5900 ± 0.0173 | 0.5133 ± 0.1493 | 0.6938 ± 0.0536 |
| | average | 0.7380 | 0.8512 | 0.7433 | 0.7193 | 0.7191 | 0.8739 | 0.6344 | 0.5838 | 0.5790 | 0.7158 ± 0.0308 |

Table 7: Single-Task Experiments on OPT-1.3B with 1000 samples. Values correspond to mean across three random seeds. FO: First-Order. FT: full-model fine-tuning.

Table 8 demonstrates the results for fine-tuning Llama2-7b [52] on various classification and open-ended generation tasks.

### D.2.1  Compare with the two-stage pipeline

To also validate that the improved results are not because of tuning more parameters, we conducted an experiment on COPA using OPT1.3B and compared Bilevel-ZOFO to a two-stage pipeline that tunes the same number of parameters. First, we performed first-order prompt tuning for a fixed number of steps (same as the number of lower-level updates in bilevel-ZOFO), followed by additional tuning using ZO for the same number of iterations as the upper level updates in bilevel-ZOFO (**A two-stage pipeline**). As shown in Table 9, even with extensive hyperparameter tuning, the second stage does not improve the results achieved after the first stage and is highly likely to decrease performance. Our method, however, improves performance when using the same number of steps in the upper and lower levels, respectively. The bilevel structure makes the trained prompts dynamically optimal for the full ZO fine-tuning and reaches an accuracy of 76.33.

| Trainer | Mode | BoolQ | ReCoRD | SQuAD | SST2 | Average |
|---|---|---|---|---|---|---|
| MeZO | ft | 0.7915 ± 0.0516 | 0.7890 ± 0.0001 | 0.7737 ± 0.1634 | 0.8646 ± 0.0216 | 0.8047 |
| | lora | 0.8020 ± 0.0014 | 0.7970 ± 0.0001 | 0.7412 ± 0.0013 | 0.8529 ± 0.0117 | 0.7983 |
| | prefix | 0.7830 ± 0.0131 | 0.7905 ± 0.0007 | 0.7093 ± 0.0207 | 0.8364 ± 0.0010 | 0.7798 |
| | prompt | 0.7787 ± 0.0049 | 0.7935 ± 0.0007 | 0.7014 ± 0.0451 | 0.8246 ± 0.0216 | 0.7746 |
| FO | lora | 0.8420 ± 0.0104 | 0.7920 ± 0.0053 | 0.8197 ± 0.0043 | 0.9557 ± 0.0007 | 0.8524 |
| | prefix | 0.7783 ± 0.0021 | 0.8013 ± 0.0012 | 0.7946 ± 0.0419 | 0.9243 ± 0.0053 | 0.8246 |
| | prompt | 0.8083 ± 0.0142 | 0.8023 ± 0.0074 | 0.7805 ± 0.0633 | 0.9284 ± 0.0072 | 0.8299 |
| Ours | lora | 0.8473 ± 0.0025 | 0.8290 ± 0.0044 | 0.8160 ± 0.0041 | 0.9629 ± 0.0053 | 0.8638 |
| | prefix | 0.8193 ± 0.0127 | 0.8067 ± 0.0065 | 0.8090 ± 0.0302 | 0.9382 ± 0.0064 | 0.8433 |
| | prompt | 0.8145 ± 0.0012 | 0.8108 ± 0.0065 | 0.7960 ± 0.0028 | 0.9222 ± 0.0039 | 0.8359 |

Table 8: Single-Task Experiments on Llama2-7B with 1000 samples. Values correspond to mean and std across three random seeds. FO: First-Order. FT: full-model fine-tuning

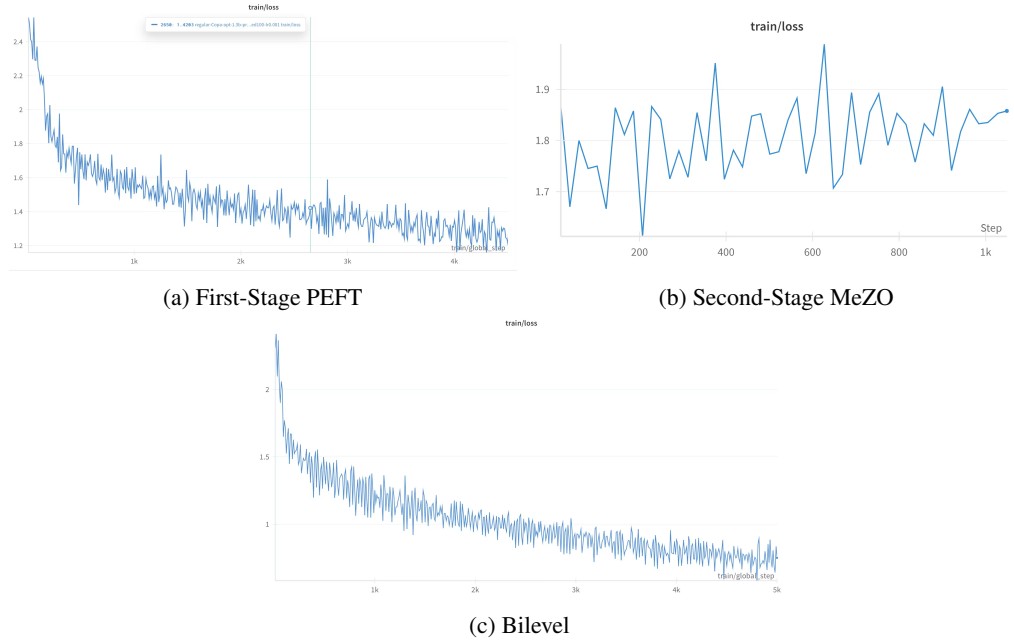

(a) First-Stage PEFT          (b) Second-Stage MeZO

(c) Bilevel

Figure 6: The training loss curves for both stages of a two-stage approach (a and b) and our bilevel framework (c).

The observed performance drop after the second stage is indeed counter-intuitive at first glance. However, it is a limitation of MeZO as it approximates gradients. While further fine-tuning intuitively should improve performance, the inherent noise in gradient approximation can lead to suboptimal updates. This observation is consistent with the fact that MeZO typically requires a significant number of iterations to converge. This is a key contribution of our work: Our approach addresses MeZO's challenges, such as sensitivity to hard prompts and long convergence times, while outperforming both MeZO and PEFT and maintaining similar memory efficiency. The intuition behind why our method is effective in enhancing both MeZO's full-model tuning and PEFT is in the nested bilevel structure. This structure encodes more information (as reflected in the training method) from the prompt tuning stage than only treating it as a first stage, thereby providing better guidance for MeZO. In contrast, our bilevel method effectively addresses the issues of MeZO and demonstrates improved performance over both MeZO and the PEFT baseline, even with the same number of ZO iterations. This phenomenon that a bilevel-method is better than a two-staged pipeline is also observed in the later work on diffusion models [49].

| Method | Experiment (COPA) | Acc (%) |
|---|---|---|
| | After Stage 1 | 74.33 |
| | After Stage 2 (lr 0.001) | 51.66 |
| Two-Stage | After Stage 2 (lr 0.0001) | 70.33 |
| | After Stage 2 (lr 0.00001) | 72.66 |
| | After Stage 2 (lr 0.000001) | 74.33 |
| Bilevel-ZOFO | - | **76.33** |

Table 9: Comparison of Bilevel-ZOFO with a two-staged pipeline.

The training loss curves for both stages of a two-stage approach and our bilevel framework are provided in Figure 6. When running MeZO in the second stage, the training loss exhibits oscillations and does not show improvement within 500–1000 iterations. This behavior is consistent with findings in the original MeZO [35] paper, which notes that MeZO typically requires much longer to converge—on the order of 100k iterations. The oscillatory behavior observed within the shorter training duration is not surprising due to gradient approximation errors.

### D.3 Multi-task experiments

In this section we explain the experimental details of mutil-task experiments.

### D.3.1 Meta-Tasks

Following the methodology of Min et al. [36], we evaluate the performance of bilevel-ZOFO as a fast and efficient meta-learning algorithm. We perform experiments using four of the distinct meta-learning settings outlined in MetaICL [36]: classification-to-classification, non-classification-to-classification, QA-to-QA, and non-QA-to-QA. Each of these *meta-learning tasks* includes a set of training sub-tasks and a different set of test sub-tasks. The sub-tasks are sourced from CROSSFIT [57] and UNIFIEDQA [19], comprising a total of 142 unique sub-tasks. These sub-tasks cover a variety of problems, including text classification, question answering, and natural language understanding, all in English. Table 10 shows the number of tasks in each training and testing meta-learning setting and the total number of examples in each training task.

| Meta-train Setting | # tasks | # examples | Target Setting | # tasks |
|---|---|---|---|---|
| Classification | 43 | 384,022 | Classification | 20 |
| Non-Classification | 37 | 368,768 | | |
| QA | 37 | 486,143 | QA | 22 |
| Non-QA | 33 | 521,342 | | |

Table 10: Details of four different meta-learning settings. Each row indicates meta-training/target tasks for each setting. There is no overlap between the training and test tasks.

See Tables 14 and 15 of MetaICL [36] for a list of all sub-tasks.

### D.3.2 Baselines

We use GPT2-Large [42] as the base model for these experiments.We compare our method against several baseline approaches:

- **MetaICL** [36]: A method for meta-learning with in-context learning. MetaICL tunes all the parameters of the base model using the first-order method. In both training and testing, the model is given $k$ demonstration examples, $(a_1, b_1), \ldots, (a_k, b_k)$, where $b_i$ represents either classification labels or possible answers in question-answering tasks, along with one test example $(a, b)$. The input is formed by concatenating the demonstration examples $a_1, b_1, \ldots, a_k, b_k, a$. The model then computes the conditional probability of each label, and the label with the highest probability is selected as the prediction.

- **Zero-shot**: This method uses the pretrained language model (LM) without any tuning, performing zero-shot inference without any demonstration examples.

- **In-context Learning (ICL)**: This method uses the pretrained LM with in-context learning by conditioning on a concatenation of $k$ demonstration examples and 1 actual test sample similar to MetaICL.

We sample 768 examples from each training sub-task. We use these samples to train MetaICL in their original setting for 30,000 steps. This includes learning rate of $1e-5$, batch size of 1 on 8 GPUs, 8-bit Adam optimizer and fp16 half precision. See MetaICL[36] for full details. To train our method, we split the training dataset of each sub-task to two subsets, 256 samples as the development dataset for upper-level updates and 512 samples for lower-level training. For each outer iteration of our method, we randomly sample a subset of 5 training tasks. We perform 10 lower-level updates between each pair of upper-level updates. To keep bilevel-ZOFO as lightweight as possible, unlike MetaICL, we do not include demonstration examples in the inputs. Since bilevel-ZOFO uses significantly less memory and has much faster updates compared to MetaICL, theoretically we are able to train it for many more iterations within the same total training duration as MetaICL. However, due to resource constraints, we only train bilevel-ZOFO for 50,000 iterations. Similar to [35], we did not observe a plateau in performance for bilevel-ZOFO, indicating that further training can yield additional improvements. We use Adam optimizer and a learning rate of $1e-6$ for both upper and lower-level training. We employ a batch size of 4 and train on a single rtx6000ada GPU.

For both ICL and MetaICL, during the testing phase the model is given $k = 4$ demonstration examples for each test data point. We don't use demonstration examples in test samples for bilevel-ZOFO evaluation. We evaluate the zero-shot capabilities of our method as well as the performance of the

final model LoRA-tuned for 10 additional iterations on 4 demonstration samples from each class of each test sub-task. Similar to [36], we report **Macro-averaged F1** as the evaluation metric.

### D.4 Additional Experiments and Clarifications

#### D.4.1 Full First-Order Fine-Tuning (FO-FT) Baselines

Our problem setting targets memory/throughput-constrained fine-tuning where full first-order (FO) updates on *all* backbone parameters are often impractical. Accordingly, the main paper emphasizes comparisons against PEFT and ZO methods that are actually deployable under such constraints. Here we include FO full finetuning results on identical data splits and evaluation protocols as our single-task experiments to contextualize the gap in Table 11 and Table 12.

Table 11: FO-FT on OPT-1.3B (accuracy).

| Method | BoolQ | CB | COPA | ReCoRD | RTE | SST-2 | WiC | WinoGrande | WSC |
|--------|-------|-----|------|--------|-----|-------|-----|------------|-----|
| FO-FT | 0.660 | 0.821 | 0.730 | 0.719 | 0.690 | 0.937 | 0.586 | 0.526 | 0.635 |

Table 12: FO-FT on Llama-2-7B (accuracy).

| Method | BoolQ | ReCoRD | SQuAD | SST-2 |
|--------|-------|--------|-------|-------|
| FO-FT | 0.863 | 0.814 | 0.801 | 0.952 |

### D.5 On Task Choice (Math/Code vs. Other Benchmarks)

We acknowledge the community's current emphasis on mathematical reasoning and code generation, but real-world fine-tuning spans QA, classification, retrieval-augmented workflows, and recommendation. These practical settings still require task-specific instructions and remain sensitive to prompt formats. To demonstrate that our observations about MeZO's prompt sensitivity also hold on popular reasoning tasks, we include a GSM8K study in Table 13

Table 13: Prompt sensitivity on GSM8K (accuracy).

| Method | With prompt ("Question:{Q}\nAnswer:{A}") | Raw format |
|--------|-------------------------------------------|------------|
| MeZO | 0.329 | 0.122 |
| Bilevel-ZOFO | 0.762 | 0.744 |

### D.6 Code & Math Experiments

We conduct code/math experiments to demonstrate transfer beyond small classification suites. Training details: Qwen2-7B is trained on GSM8K (train), HumanEval (train), and Math500 (4:1 train/test). It is evaluated on the standard test splits. LoRA and Bilevel-ZOFO are trained for 2000 steps. MeZO is trained for 10000 steps. Metrics are accuracy (GSM8K, Math500) and pass@1 (HumanEval). Across GSM8K, Math500, and HumanEval, Bilevel-ZOFO consistently improves over LoRA and strongly over MeZO.

### D.7 Other PEFT Variants

Bilevel-ZOFO is a *framework*: the lower level can adopt any FO-PEFT method and the upper level any ZO estimator. To illustrate compatibility beyond LoRA-style adapters, we add results with DePT [47], a prompt-tuning method in Table 15. We see gains persist when swapping in a stronger PEFT variant.

### D.8 More Realistic Applications

We showed the applicability of bilevel-zofo in Meta Learning. Future work can explore bilevel-zofo in Multi-Task Reinforcement Learning to tune an LLM on multiple domains. Also another application

Table 14: Qwen2-7B on math/code tasks.

| Method | GSM8K (acc) | Math500 (acc) | HumanEval (pass@1) |
|---|---|---|---|
| Before tuning | 0.420 | 0.18 | 0.476 |
| LoRA | 0.727 | 0.28 | 0.518 |
| MeZO | 0.329 | 0.05 | 0.110 |
| Bilevel-ZOFO (ours) | **0.762** | **0.31** | **0.543** |

Table 15: Llama-2-7B with DePT (accuracy).

| Method | BoolQ | SST-2 |
|---|---|---|
| DePT | 0.813 | 0.932 |
| MeZO | 0.792 | 0.865 |
| Bilevel-ZOFO + DePT | **0.852** | **0.946** |

of bilevel-zofo is in Federated or privacy-sensitive scenarios where clients can run small FO-PEFT steps locally and aggregate ZO signals centrally, which we leave to future work.

