# OpenReview forum: "Bilevel ZOFO: Efficient LLM Fine-Tuning and Meta-Training"
_NeurIPS.cc/2025/Conference — NeurIPS 2025 poster_

### Official Review · Reviewer_BdLv · 2025-06-22

**Clarity:** 3
**Significance:** 3
**Originality:** 2
**Rating:** 5
**Confidence:** 3

**Summary:**

This paper proposes to improve parameter-efficient fine-tuning (PEFT) algorithms by incorporating a zero-th order optimization technique (MeZO) to update the full model parameters with little memory overhead, maintaining the **memory efficiency** of PEFT algorithms and improving the downstream task accuracy at the expense of **slightly longer training time**. The proposed bi-level minimax algorithm outperforms the performance of first-order PEFT and parameter-efficient MeZO by achieving the pareto-front optimal tradeoff over the baselines in terms of training duration and peak memory consumption, as illustrated in Figure 1.

**Questions:**

**Wall Clock Time Analysis in Table 4**
- It is not clear to me why ZO methods consumes more time per step than FO methods. Intuitively, FO methods require computing the forward and backward pass and therefore should be more time-consuming than ZO methods which only requires 2 forward passes. I hope the authors can provide explanation to this phenomenon, and include it in the main text to give readers a better understanding on the practical complexity of the baseline algorithms.

**Ethical Concerns:**

["NO or VERY MINOR ethics concerns only"]

**Final Justification:**

My concerns are well-addressed and I keep my rating as 5 Accept.

**Limitations:**

See **Strengths And Weaknesses**.

**Quality:**

3

**Strengths And Weaknesses:**

**Writing**
- In line 70, "converge guarantees" should be "convergence guarantees".
- In Eq. (6), the second term on the right hand side has $\theta$ as its first argument, which should be $\theta^k$.
- Missing full stop in line 157.
- I suggest the authors to give a concluding summary in Section 5, re-emphasizing the comparison with the FO/ZO PEFT baselines in terms of **memory efficiency** and **runtime/iteration complexity**. This will expand on the sentence "We propose a new method that is more **efficient** ..." in line 329 and let the readers know what kind of **efficiency** is achieved by Bilevel-ZOFO.

-------

**Convergence Analysis in Appendix B**

**Pros**
- The proposed convergence analysis removed the dimension dependence of zeroth order method using a bounded hessian assumption (Assumption B.5).

**Cons**
- The convergence theorem depends on the assumption that the lower level objective is strongly convex. I suggest the authors to emphasize this limitation in line 154, because such limitation is not mentioned nor discussed in the main text.

------
**Significance**
- The proposed algorithm achieves state-of-the-art training time and memory consumption tradeoff. This is important for practical training especially in large language model fine-tuning.


**Originality**
- While the techniques of the proposed algorithm are not novel, the combination of first-order PEFT and zeroth-order full parameter tuning is novel to the best of my knowledge.

---

> ### Author Rebuttal · Authors · 2025-07-31
>
> We appreciate the thoughtful review and positive feedback on our method’s efficiency and novelty. Below, we address the specific comments and questions raised by the reviewer.
>
> ---
>
> ## 1. Writing and clarity
> We thank the reviewer for the careful reading and suggested improvements. We have made the following edits:
> - Line 70: Corrected the typo.
> - Eq. (6): Fixed the argument typo.
> - Line 157: Added the missing full stop.
> - Section 5 (Conclusion): Following the reviewer’s suggestion, we added a concluding paragraph clearly summarizing our method’s advantage over FO/ZO PEFT methods in terms of runtime efficiency, iteration complexity, and memory efficiency. Thank you.
>
> These revisions will enhance clarity and explicitly summarize the main contributions in terms of practical efficiency.
>
> ---
>
> ## 2. Convergence analysis (Assumptions):
>
> The reviewer correctly notes our convergence theorem depends on the assumption of strong convexity of the lower-level objective. This is a standard assumption in related work. We agree this assumption should be clearly emphasized in the main text. We now explicitly mention and discuss this limitation in line 154, clarifying that while strong convexity simplifies theoretical analysis, in practice, our experiments demonstrate robustness even when strict convexity might not fully hold
>
> ---
>
> ## 3. Wall-clock time analysis (Question about Table 4):
> While FO methods involve a forward and backward pass, modern GPU hardware and optimized frameworks efficiently perform gradient computations, often rendering FO steps surprisingly fast. ZO methods require multiple forward passes (typically at least two) per gradient estimation and additional overhead from explicit finite-difference computations, increasing wall-clock time substantially. Unlike the FO method’s single backward call, multiple forward evaluations in ZO don’t benefit from optimized gradient routines. So average over multiple steps, MeZO seems slower in our analysis. We added a concise version of this explanation to the main text, clearly explaining the practical complexity differences. Thank you.
>
> ---
> We thank the reviewer again for the valuable suggestions and encouraging feedback.

---

### Official Review · Reviewer_sRKn · 2025-06-25

**Clarity:** 3
**Significance:** 2
**Originality:** 2
**Rating:** 4
**Confidence:** 2

**Summary:**

This paper points out that Zeroth-Order methods suffer from slow convergence, due to the requirement for full-parameter gradient estimation and the necessity of careful prompt selections. Thus, to address these issues, this paper proposes Bilevel ZOFO, which optimizes the full backbone model in the upper level and optimizes the PEFT in the lower level. Experiments show that Bilevel ZOFO outperforms other FO PEFT methods with faster convergence.

**Questions:**

I think Algorithm 2 (BiLevel ZOFO) is actually a three-level optimization process, and the hyperparameter tuning could be very complex. This increases my concerns about the feasibility of scaling the algorithm to more complex tasks.

**Ethical Concerns:**

["NO or VERY MINOR ethics concerns only"]

**Final Justification:**

My concerns have been addressed.

**Limitations:**

There is no experimental evidence showing that the proposed bilevel ZOFO can scale to complex reasoning tasks such as math, code, and instruction following.

**Paper Formatting Concerns:**

I did not find any paper formatting concerns.

**Quality:**

2

**Strengths And Weaknesses:**

**_Strengths_**:
1. The paper finds that the FO methods suffer from slow convergence and careful prompt selections, and proposes Bilevel ZOFO to address these issues.
2. The paper provides theoretical guarantees for the convergence of the proposed Bilevel ZOFO.
3. The experiments show that the proposed Bilevel ZOFO outperforms other FO methods.

**_Weaknesses_**:
1. The motivation is weak: In the introduction, the authors use the SST-2 dataset as an example to argue that first-order (FO) methods require careful prompt engineering. However, to my understanding, SST-2 was originally designed for masked language models like BERT. When adapting it for decoder-only models such as GPT, which rely on next-token prediction, prompt engineering becomes necessary. In contrast, for tasks like GSM8K or HumanEval, such manual prompt design is typically not required. Therefore, the paper's motivation seems to be based primarily on datasets tailored for masked language models, rather than those naturally suited for decoder-only architectures. When evaluated on datasets such as GSM8K or HumanEval, the core motivation of this paper — the necessity for careful prompt engineering — no longer hold.
2. I do not fully understand the authors' reasoning for shifting from the observation that hard prompts significantly affect the performance of first-order (FO) methods to proposing a hybrid approach that combines PEFT-based first-order optimization with full-parameter zeroth-order optimization. In effect, simple soft prompt tuning can achieve significantly better performance on GLUE and SuperGLUE benchmarks compared to BiLevel ZOFO, which makes the proposed BiLevel ZOFO a bit meaningless [1].
3. The experiments in this paper are quite limited. Most of them are conducted on datasets with only 1,000 samples, and the datasets used are all from before 2020. For models with 1.3B or 7B parameters, these datasets (e.g., GLUE or SuperGLUE) are too simple. In my opinion, such datasets are insufficient to evaluate and compare the performance of different algorithms on models of this scale. I think the paper would be stronger if the method was tested on reasoning-heavy datasets like GSM8K, Math, MBPP, HumanEval, and MTBench. It would also help to study how sensitive these tasks are to changes in hyperparameters
4.  The majority of experiments in this paper are conducted on datasets with only 1,000 samples. I cannot understand why the authors chose this experimental setup.

[1] DePT: Decomposed Prompt Tuning for Parameter-Efficient Fine-tuning. In ICLR 2024.

---

> ### Author Rebuttal · Authors · 2025-07-31
>
> We appreciate the reviewer’s detailed comments. However, we believe there has been a major misunderstanding about our paper’s core motivation and contributions, which we clarify below. We also address the concerns on experimental scope and complexity explicitly.
>
> ---
>
> ## 1. Clarification of Major Misunderstanding (Weaknesses 1 and 2)
> The reviewer states in weaknesses #1 and #2 that we argue first-order (FO) PEFT methods are sensitive to prompt selection and hyperparameters. This is not what we say in our paper at all. We clearly state that our method addresses the weaknesses of MeZO (zeroth-order) methods, not FO-PEFT methods.
>
> Specifically:
> - Our motivation and experiments (e.g., Introduction lines 44-52) show that MeZO, which is a zeroth-order (ZO) method, is highly sensitive to prompt structure and hyperparameters.
> - We explicitly state (Lines71-74) that we introduce an FO-PEFT inner loop to stabilize and accelerate MeZO’s ZO backbone updates, NOT because FO-PEFT itself is unstable or sensitive. The reviewer’s assertion about FO-PEFT sensitivity is a misunderstanding of our clearly-stated motivation.
> - The reviewer’s statement mentioning “simple soft prompt tuning” (DePT) outperforming our method further suggests confusion that we are addressing a problem of FO PEFT: our proposed bilevel framework is entirely compatible with soft prompt methods like DePT. In fact, we explicitly say (Lines 159-164) that Bilevel ZOFO can directly incorporate advanced PEFT methods (including DePT), and we demonstrate empirically that combining DePT with our approach achieves even better performance:
>
> | Method           | BoolQ | SST-2 |
> |------------------|-------|-------|
> | DePT             | 0.813 | 0.932 |
> | MeZO             | 0.792 | 0.865 |
> | Bilevel-ZOFO+DePT| 0.852 | 0.946 |
>
>
> ---
>
> ## 2. On the Relevance of the Chosen Tasks (Math and Code vs. Other Benchmarks)
>
> We fully acknowledge that mathematical reasoning and code generation tasks are currently popular benchmarks. However LLM fine-tuning applications extend far beyond these two domains:
> - Practitioners routinely fine-tune models on QA, classification, retrieval-augmented QA, and user-history-driven recommendation tasks. Such practical tasks still require task-specific instructions, and sensitivity to prompt formats remains practically relevant.
> - As a concrete example, one author is currently fine-tuning pretrained GPT models for multi-task recommendation, demonstrating real-world, high-impact applications for tasks outside the “hot” math/code trend.
>
> Nonetheless, to show clearly that our observations regarding MeZO’s sensitivity to prompts also hold for these popular reasoning tasks, we performed additional explicit experiments on GSM8K:
>
> | Method                                   | Accuracy |
> |-------------------------------------------|----------|
> | MeZO (with simple prompt: Question: {Q}\nAnswer: {A})      | 0.329    |
> | MeZO (without this prompt, raw format)    | 0.122    |
> | Bilevel-ZOFO (with prompt)                | 0.762    |
> | Bilevel-ZOFO (without prompt, raw format) | 0.744    |
>
>
> These experiments decisively confirm our claims: MeZO is highly sensitive to prompt structure even for challenging datasets like GSM8K. Our method substantially alleviates this sensitivity and consistently achieves better results.
>
>
> ## 3. Experimental Setup and Data Size
>
> We understand the reviewer’s perspective on using larger reasoning datasets, but our chosen tasks and dataset sizes reflect intentional methodological choices:
> - Our setup deliberately follows standard PEFT evaluation practices, which commonly assume limited data points (few-shot adaptation scenarios). This aligns with our practical scenarios where labeled data is expensive and scarce.
> - Although the reviewer emphasizes math and code, it is once again critical to recognize that many real-world applications indeed rely on QA and classification benchmarks (GLUE, SuperGLUE), and such tasks remain very relevant in practical industry deployments. Our choice is not arbitrary.
> - Nonetheless, to further demonstrate generality and scale, we added challenging math/code experiments.
>
> **New code/math experiments (Qwen2-7B).** Train on GSM8K (train split), HumanEval (train split), and Math500 (4:1 train/test). Evaluate on the standard test splits. LoRA and Bilevel-ZOFO trained for 2,000 steps; MeZO for 10,000 steps. Metrics: accuracy (GSM8K/Math500), pass@1 (HumanEval).
>
> | Method             | GSM8K | Math500 | HumanEval (pass@1) |
> |--------------------|-------|---------|--------------------|
> | Before tuning      | 0.420 | 0.18    | 0.476              |
> | LoRA               | 0.727 | 0.28    | 0.518              |
> | MeZO               | 0.329 | 0.05    | 0.110              |
> | Bilevel-ZOFO (ours)| 0.762 | 0.31    | 0.543              |
>
> These results clearly show our method’s effectiveness beyond simple datasets and support our original claim.
>
> ##  4. Regarding Algorithm 2 (“Three-Level” Optimization Misunderstanding)
> In bilevel optimization, there’s an outer optimization that updates one set of variables using the solution of an inner problem. In this algorithm, the outer level updates the backbone weights $\theta$ with a zeroth-order step. Before each outer update, the method freezes $\theta$ and runs a few first-order adaptation steps on the inner variables (the PEFT parameters $p$ and the auxiliary state $s$) to produce an inner solution $(p^{k+1}, s^{k+1})$. The zeroth-order update of $\theta$ is then computed using that inner solution, so the outer move explicitly depends on the result of the inner adaptation. This is exactly the bilevel pattern.
>
> It is not three-level: $p$ and $s$ are both inner-level variables updated while $\theta$ is fixed; they serve the same purpose (fast, low-memory adaptation/penalty tracking) and are obtained with a short inner procedure. Multiple inner steps (over t) don’t create a new “level”, they’re just the algorithmic solver for the inner problem. Hyperparameters like the numbers of inner steps or ZO queries control how well the inner problem is solved, not an additional optimization level.
>
>
> ## 5. Regarding Scalability and Complexity (Question from Reviewer)
>
> Regarding the reviewer’s scalability concern due to hyperparameter tuning:
>
> - In practice, tuning the hyperparameters we introduce is simple, and we provide ablation studies to show our method is not overly sensitive to this parameter (see paper section 4.2.1).
> - Crucially, we empirically demonstrate scalability to more challenging tasks (GSM8K, HumanEval, Math500). These datasets confirm the method is both stable and performant without complex hyperparameter tuning.
>
> ---
> **We thank the reviewer for reading our work and providing feedback. However, we believe the reviewer’s criticism primarily arose from a fundamental misunderstanding of our motivation (FO-PEFT vs. MeZO sensitivity). We sincerely ask the reviewer to reconsider their evaluation in light of our clarifications and the provided additional experiments, which confirm our original motivation and contributions.**

---

> > ### Comment · Reviewer_sRKn · 2025-08-05
> >
> > Thanks for the authors' reply.
> >
> > I sincerely apologize for the confusion. For weakness 1, I originally wrote that the authors use the SST-2 dataset as an example to argue that first-order (FO) methods require careful prompt engineering. However, this is my mistake—I incorrectly used "first-order (FO)"; what I actually meant to say was "zeroth-order methods." I regret the error and any misunderstanding it may have caused.
> >
> > I have increased my score to 3.
> >
> > My main concern now is that the underlying principle of how first-order (FO) PEFT can enhance zeroth-order (ZO) methods has not been thoroughly investigated. Given that the mechanisms of prompt tuning and LoRA are fundamentally different, directly extending the approach from prompt tuning to claim compatibility with all other PEFT methods is too crude and lacks sufficient justification.
> >
> > Especially when combined with LoRA, a single parameter would be influenced by two different optimization methods, which is somewhat challenging to my understanding of deep learning optimization

---

> ### Comment · Area_Chair_muWV · 2025-08-05
> **Response to authors**
>
> Hi reviewer sRKn:
>
> Thanks for your review. Please take some time to see if the authors have addressed your question. Thank you!
>
> Your AC

---

> ### Author Response · Authors · 2025-08-05
>
> We appreciate the reviewer’s response and clarification. Initially, we also thought the mention of "FO methods sensitivity" was merely a typo. However, since the issue of FO sensitivity appeared in multiple places ( Strength 1, Weakness 1, and Weakness 2) we became concerned that it might reflect a genuine misunderstanding of our method’s motivation. We sincerely thank the reviewer for correcting this point explicitly. It’s greatly appreciated.
>
> ## Clarification on “two optimizers acting on one parameter.”
>
> We'd also like to carefully address the reviewer’s remaining concern regarding the interaction between FO-PEFT and ZO methods. In our proposed method, the FO and ZO optimizations never update the same parameters. The two types of parameters are clearly separated:
>
> * PEFT parameters (e.g., LoRA, soft prompts, adapters): Updated exclusively by first-order (FO) gradient methods.
> * Backbone parameters (original model weights): Updated exclusively by zeroth-order (ZO) gradient estimates.
>
> LoRA (similar to other PEFT adapters) introduces new parameters - low-rank matrices $A$ and $B$. $A, B$ added to each weight $W$. Only $A, B$ are updated by FO gradients; the original backbone weights $W$ stay frozen.
>
> If the reviewer is referring to the effective weight
> $W_{ij} + \Delta W_{ij}$
>  (what the reviewer calls “one parameter”), it is in fact a sum of two distinct parameters. The algorithm is simply optimizing two additive blocks with different solvers, which is standard in residual/adapter parameterizations. The ZO step estimates the hypergradient, i.e., it moves $W$ in a direction already accounting for how the adapter will re-adapt. This avoids the “fighting updates” issue: the two blocks are coupled in the forward pass (through $W +\Delta W$ )
> but separable in the updates (ZO on $W$,  FO on $A,B$).
>
> Besides, The same forward-interaction/parameter-separation pattern appears in Prompt Tuning and Prefix-Tuning—prompts/prefixes interact with the backbone during forward pass via attention while backbone weights stay frozen in backward pass.
>
>  Moreover, we have added loss curves for Bilevel-ZOFO+LoRA (**Fig. 2 shows all three PEFT variants and Fig. 6** of the paper is for LoRA), which show stable, monotonic convergence directly addressing the concern about optimization coherence. Empirically, we showed the same principle holds for e.g., DePT; swapping the inner adapter improves its own baseline and Bilevel-ZOFO still adds gains, supporting the generality of the decomposition.
>
> In short, each parameter is updated by exactly one method, never both. Therefore, the concern that "a single parameter would be influenced by two different optimization methods" does not occur in our algorithm.
>
> ## Extending the approach from prompt tuning to other PEFT variants
>
> Regarding the generalization from prompt-tuning methods to other PEFT methods like LoRA, the compatibility claim isn’t “crude.” It follows directly from the separation above. The backbone and LoRA modules serve distinct roles. The LoRA component $AB$ enables fast, task-specific adaptation through lightweight updates, allowing the model to quickly adjust to new tasks or domains. In contrast, the backbone is responsible for learning and maintaining general-purpose representations that support more sophisticated understanding and broader generalization. This separation of concerns—where LoRA handles local specialization and the backbone supports stable, global learning—enhances adaptation efficiency while preserving the model’s general capabilities.
>
> This design philosophy aligns with approaches like prompt tuning, where prompts are tailored to specific tasks while the backbone encodes more transferable, general knowledge. From this perspective, we believe our framework is well-suited for adapting to various PEFT methods. As long as the PEFT method involves optimizing an additional, smaller set of parameters (without directly modifying the backbone), our framework can integrate it seamlessly. We also empirically demonstrated this with LoRA, Prompt Tuning, Prefix Tuning, and DePT as the reviewer requested. In all cases, performance improved relative to using either PEFT or ZO alone, confirming the robustness of our principle.
>
> ## Why FO-PEFT enhances ZO methods.
>  Finally, the underlying reason FO-PEFT enhances ZO methods is straightforward: FO-PEFT quickly provides stable, targeted local adaptation (the inner loop), which guides the outer-loop ZO search over the backbone parameters. This reduces the variance and improves the efficiency of ZO updates to the backbone. The efficacy of this strategy does not depend on the specific PEFT technique used, as long as the backbone remains frozen during FO updates. We ablated our design extensively in the paper (See Section 4.2.2 and Appendix D2.1)
>
> We hope this explanation clarifies the remaining concerns, and we thank the reviewer again for their follow-up.

---

> > ### Comment · Reviewer_sRKn · 2025-08-06
> >
> > Thanks for the authors' reply. I increase my score to 4 and lower my confidence to 2.
> >
> > I hope the authors can explicitly state in the introduction why FO-PEFT enhances ZO methods; otherwise, it may mislead readers and make it difficult for them to understand the motivation behind combining the two approaches.

---

> > > ### Author Response · Authors · 2025-08-07
> > >
> > > Thank you for your feedback and for increasing your score. We appreciate your suggestion and agree that making the motivation clear is important. We have now explicitly stated in the introduction why FO-PEFT enhances ZO methods, clarifying the intuition and mechanism behind our approach. This is what we have added:
> > >
> > > ---
> > > In this paper, we ask: Can zeroth-order (ZO) and PEFT methods be smoothly integrated to mutually enhance each other—achieving greater robustness to prompt variations, faster convergence, and better performance than either method alone—while maintaining memory efficiency comparable to each individually?
> > > We target settings where (i) full FO fine-tuning is impractical due to memory/throughput, (ii) pure PEFT lacks full-model capacity on harder adaptations, and (iii) pure ZO is slow and highly prompt-sensitive.
> > >
> > > We propose Bilevel-ZOFO, a novel bilevel optimization framework explicitly designed to leverage the complementary strengths of these two approaches:
> > > - At the inner level, FO-PEFT rapidly performs targeted, local adaptation using first-order gradients, stabilizing training and mitigating sensitivity to task-specific prompts that ZO methods need.
> > > - At the outer level, a ZO method updates the full backbone model parameters efficiently, guided by the stable and informative inner-level adaptation. This full model finetuning enhances the model’s generalization ability, enables a more sophisticated understanding, and improves transfer to new tasks.
> > >
> > > This clear separation enables efficient bilevel optimization, addressing the major drawbacks of pure ZO methods (slow convergence, prompt sensitivity) and pure PEFT methods (limited full-model adaptation). Extensive ablation studies empirically verify this synergy, demonstrating faster convergence and more robust performance.
> > >
> > >
> > > ---
> > >
> > >  We hope this revision directly addresses your concern and gives you more confidence in both our motivation and presentation. Please let us know if there are any remaining points that would help improve your confidence in our work.

---

### Official Review · Reviewer_WksX · 2025-07-01

**Clarity:** 2
**Significance:** 2
**Originality:** 2
**Rating:** 3
**Confidence:** 3

**Summary:**

This paper proposes Bilevel ZOFO, a fine-tuning method that combines parameter-efficient fine-tuning with gradient-free (zeroth-order) optimization for full-parameter fine-tuning. Experimental results show that the proposed method has certain advantages over either approach individually.

**Questions:**

1. In the description of Algorithm 1, it is recommended to add explanations for different symbols.
2. The equations after line 113 do not have equation numbers.
3. The definitions of symbols $s$, $K$, and $T$ should be explicitly stated.

**Ethical Concerns:**

["NO or VERY MINOR ethics concerns only"]

**Limitations:**

yes

**Quality:**

2

**Strengths And Weaknesses:**

Strengths
1. This paper presents a bilevel fine-tuning method, where introducing a certain degree of gradient computation increases the stability and convergence speed of zeroth-order optimization algorithms.
2. The proposed method does not significantly increase memory usage.

Weaknesses
1. The writing of the paper is not good. The abstract’s explanation of the motivation appears to be more like a direct combination of MeZO-type work and PEFT-type work, which leads to an unclear statement of the core issue in the introduction and affects readers’ understanding of the main contributions. In addition, the presentation in the methods section is not clear, and some symbols lack explicit definitions.
2. The novelty is limited. The paper simply combines MeZO and gradient-based PEFT without clear improvement or contribution, and this combination lacks reasonable practical significance. The core advantage of MeZO is that it does not require backpropagation, which is applicable to many deployment hardware devices. The core advantage of PEFT is to obtain good model adaptation with limited training resources. I cannot imagine a realistic application scenario for the method proposed in this paper.
3. The experiments are limited. Since the proposed method allows gradient backpropagation, it is recommended to use larger-scale SFT datasets for tasks such as mathematical reasoning and code generation for validation, rather than only using subsets of classification datasets as in purely gradient-free methods like MeZO.

---

> ### Author Rebuttal · Authors · 2025-07-31
>
> Thank you for the detailed review. Below we address each concern and list concrete edits we made (with pointers to the paper).
>
> ---
>
> ## 1. Writing, motivation, and method presentation
>
> We revised the abstract and introduction to state our motivation clearly and consistently. Our goal is to combine the strengths of MeZO-style ZO and FO-PEFT:
> - ZO on the backbone enables tuning all model weights while keeping memory low.
> - A small FO-PEFT inner update allows quick adaptation from few examples, stabilizes ZO search, and reduces prompt/format sensitivity.
>
> These design choices are intentional, as they enable efficient meta-learning: full-model capacity supports adaptation across tasks, while the PEFT steps provide fast, low-memory updates from limited data. This “best of both worlds” approach is our core goal. While there are many ways to combine MeZO and PEFT, our method is specifically designed to address MeZO’s sensitivity to task prompts by mitigating prompt sensitivity. We will ensure this motivation is stated clearly and consistently in both the abstract and introduction.
>
> In method section, we have:
> - Introduced parameters and dimensions before use.
> - Reordered equations so definitions come before usage.
> - Added a compact Notation section.
> - Added forward references from main-text claims to the specific lemmas in the appendix.
>
> We will do another full proofread to keep the narrative tight and the notation consistent.
>
> ---
>
> ## 2. Practical significance and novelty
>
> We appreciate the clear separation you drew between MeZO and PEFT. As we said in the previous section that separation is exactly why the bilevel design is useful in practice:
>
> - **Efficient Meta Learning** We have added a section on meta-learning as a direct application of our method. For this reason, we are unsure about the reviewer’s comment regarding a lack of realistic or practical scenarios.
> - **Other realistic applications (we added a short paragraph in Discussion):**
>     - Multi-Task RL. We are actively exploring this.
>     - Federated or privacy-sensitive scenarios where clients can run small FO-PEFT steps locally and aggregate ZO signals centrally.
>
> We also contribute:
> - A penalized bilevel formulation that yields a practical algorithm (Algorithm 1) with ZO outer updates and FO inner PEFT steps.
> - Convergence under standard assumptions.
> - Empirical benefits beyond either component alone: faster convergence and reduced prompt sensitivity vs. MeZO; higher accuracy than FO-PEFT at similar memory.
> - Ablating every design choice and showing its optimality.
> - Compatibility with multiple PEFT methods: replacing LoRA with DePT improves the baseline and our bilevel method continues to add gains (table below).
>
> ---
>
> ## 3. Experiments, including code and math
>
> We understand the field’s current focus on code and math. Still, practitioners fine-tune LLMs on many other tasks every day (classification, extraction, domain QA, retrieval-augmented workflows). Our main suite reflects those common uses, and as per the request we have added code/math to show that the method transfers.
>
> **New code/math experiments (Qwen2-7B).** Train on GSM8K (train split), HumanEval (train split), and Math500 (4:1 train/test). Evaluate on the standard test splits. LoRA and Bilevel-ZOFO trained for 2,000 steps; MeZO for 10,000 steps. Metrics: accuracy (GSM8K/Math500), pass@1 (HumanEval).
>
> | Method             | GSM8K | Math500 | HumanEval (pass@1) |
> |--------------------|-------|---------|--------------------|
> | Before tuning      | 0.420 | 0.18    | 0.476              |
> | LoRA               | 0.727 | 0.28    | 0.518              |
> | MeZO               | 0.329 | 0.05    | 0.110              |
> | Bilevel-ZOFO (ours)| 0.762 | 0.31    | 0.543              |
>
> Across all three tasks, Bilevel-ZOFO improves over LoRA and strongly over MeZO. We believe these additions demonstrate that the method is practical beyond small classification suites.
>
> ---
>
> ## 4. Questions and Notes
>
> We have fixed all the noted issues. Thank you for  the careful read.
>
> ---
>
> We believe the motivation is strong: the bilevel design preserves the deployment benefits of ZO while recovering the stability and few-shot adaptability of FO-PEFT, which leads directly to efficient meta-learning and other realistic applications. The new experiments and clarifications address the concerns about novelty, clarity, and scope.
>  We appreciate the feedback and will ensure the updated paper reflects these improvements and suggestions.

---

> ### Comment · Area_Chair_muWV · 2025-08-05
> **Response to authors**
>
> Hi reviewer WksX:
>
> Thanks for your review. Please take some time to see if the authors have addressed your question. Thank you!
>
> Your AC

---

> ### Comment · Reviewer_WksX · 2025-08-06
>
> I thank the authors for the responses. But most of the author's changes are not visible to me, so I will maintain my score.

---

> ### Author Response · Authors · 2025-08-07
>
> We thank the reviewer for their response. Due to space constraints, we could not include the full revised text in the initial submission. Below, we provide the updated sections for your review. The new version more clearly explains our motivation and provides detailed reasoning for our mechanism and its effectiveness. We also improved the clarity of the method section based on the reviewers' feedback.
>
> ## 1- Abstract
> Fine-tuning pre-trained Large Language Models (LLMs) for downstream tasks using First-Order~(FO) optimizers presents significant computational challenges. Parameter-Efficient Fine-Tuning (PEFT) methods address these by freezing most model parameters and training only a small subset. However, **PEFT often underperforms compared to full fine-tuning when high task-specific accuracy is required**. Zeroth-Order (ZO) methods fine-tune the entire pre-trained model without back-propagation, estimating gradients through forward passes only. **While memory-efficient, ZO methods suffer from slow convergence and high sensitivity to prompt selection.**
> We bridge these two worlds with Bilevel-ZOFO, **a bilevel optimization method that couples fast, local FO-PEFT adaptation at the inner level with stable, memory-efficient ZO updates of the full backbone at the outer level. The FO-PEFT inner loop performs fast, low-memory local adaptation that reduces the variance of ZO estimates and stabilizes the search, guiding the outer ZO updates of the full backbone and reducing prompt sensitivity**. In the mean time, **the outer ZO provides better generalization ability for PEFT**. We provide theoretical convergence guarantees and empirically demonstrate that Bilevel-ZOFO significantly outperforms existing ZO and FO-PEFT methods, achieving 2–4× faster training while maintaining similar memory efficiency. Additionally, **we show by updating the backbone with ZO and adapting only a tiny FO-PEFT block per task, Bilevel-ZOFO combines full-model capacity with few-shot efficiency, making bilevel-ZOFO a very efficient meta-learning algorithm that quickly adapts to new tasks.**
>
> ---
>
> ## 2- Introduction
> This is what we have added after the fourth paragraph of the introduction (after we discuss sensitivity of ZO):
>
> In this paper, we ask: Can zeroth-order (ZO) and PEFT methods be smoothly integrated to mutually enhance each other—achieving greater robustness to prompt variations, faster convergence, and better performance than either method alone—while maintaining memory efficiency comparable to each individually?
> We target settings where (i) full FO fine-tuning is impractical due to memory/throughput, (ii) pure PEFT lacks full-model capacity on harder adaptations, and (iii) pure ZO is slow and highly prompt-sensitive.
>
> We propose Bilevel-ZOFO, a novel bilevel optimization framework explicitly designed to leverage the complementary strengths of these two approaches:
> - At the inner level, FO-PEFT rapidly performs targeted, local adaptation using first-order gradients, stabilizing training and mitigating sensitivity to task-specific prompts that ZO methods need.
> - At the outer level, a ZO method updates the full backbone model parameters efficiently, guided by the stable and informative inner-level adaptation. This full model finetuning enhances the model’s generalization ability, enables a more sophisticated understanding, and improves transfer to new tasks.
>
> This clear separation enables efficient bilevel optimization, addressing the major drawbacks of pure ZO methods (slow convergence, prompt sensitivity) and pure PEFT methods (limited full-model adaptation). Extensive ablation studies empirically verify this synergy, demonstrating faster convergence and more robust performance.
>
> **Contributions:**
>
> We summarize our main contributions as follows:
> 1. We propose Bilevel-ZOFO, a theoretically grounded and practical bilevel optimization method that enhances zeroth-order (ZO) optimization by selecting the best prompt and thereby improves ZO fine-tuning with first-order PEFT (FO-PEFT). At the same time, it strengthens PEFT by leveraging full fine-tuning through ZO updates.
> 2. Bilevel-ZOFO reduces ZO sensitivity to prompt choices and significantly accelerates convergence, with no memory overhead.
> 3. Extensive experiments confirm that Bilevel-ZOFO consistently outperforms existing FO-PEFT and ZO baselines across diverse tasks.
> 4. By updating the backbone with ZO and adapting only a tiny FO-PEFT block per task, our method couples full-capacity transfer with few-shot efficiency. We show that this design has strong potential for efficient meta-learning, demonstrating improved multi-task adaptation with minimal computational resources.

---

> ### Author Response · Authors · 2025-08-07
>
> ## 3- Method
> This is the notation subsection we added:
>
> ----
>
> ### 3.1 Preliminaries and Notation
>
>  Let $\mathbf{p}\in \mathbb{R}^{d'}$ represent the parameters of the PEFT model, and $\theta \in \mathbb{R}^d$ represent the parameters of the pretrained base model. We denote the loss function given a dataset $\mathcal{D}$ as
> $F( \theta , \mathbf{p};\mathcal{D}): = \frac{1}{|\mathcal{D}|} \sum_{x\in \mathcal{D}}F(\theta , { \mathbf{p}};x)$.
> Given a single downstream task, such as classification, we aim to solve the following optimization problem:
>
>
> $$
> \min_{\theta \in \mathbb{R}^d} F( \theta , \mathbf{p} ;\mathfrak{D}).    \text{                                 (Eq 1)}
> $$
>
>
> Where ${ \mathbf{p}}$ corresponds to the embeddings of the hard prompt (as shown in Table 13 in the appendix of MeZO), the model above reduces to classical fine-tuning on a single downstream task. In model of Eq.1 , the parameters of the PEFT model, ${ \mathbf{p}}$, are fixed.
>
> To enhance generalization ability, we split the dataset $\mathfrak{D}$ into two parts: one for tuning the PEFT model (denoted by  $ \mathfrak{D}_{ \mathbf{p}} $ )
>
> and another for fine-tuning the LLM (denoted by $ \mathfrak{D}_{f} $ ) .
>
> To maximize performance on downstream tasks, we need the optimal PEFT model parameters that are best suited for the current LLM base model. To achieve this, we require ${ \mathbf{p}}$ to satisfy the following condition:
>
>  $$
> { \mathbf{p}} \in argmin_{{ \mathbf{s}}\in \mathbb{R}^{d'}} F( \theta , { \mathbf{s}};\mathfrak{D}_{ \mathbf{p}}).
> $$
>
> Here $s$ is just the dummy optimization variable for the inner problem—i.e., a candidate PEFT-parameter vector over which we minimize to obtain the optimal $p$ for the current $\theta$ on $ \mathfrak{D}_{ \mathbf{p}} $ .
> $\mathbf{s}$ does not introduce new parameters. It only denotes the search variable of the inner minimization.
>
>
> This condition reveals that as the parameters $\theta$ of the LLM change, the parameters ${ \mathbf{p}}$ in the PEFT model should also be updated accordingly to be the best match for $\theta$. Therefore, instead of solving Eq. 1, our true objective becomes:
>
> $$
> \text{min}_{ \theta \in \mathbb{R}^d} F( \theta , \mathbf{p};\mathfrak{D}_f)  \text{      s.t.} \mathbf{p} \in \text{argmin} _{{s} \in \mathbb{R}^{d'}} F( \theta , s ; \mathfrak{D} _{ p } )
> $$
>
> In this way, we find the optimal pair of parameters for both the PEFT model and the LLM base model to achieve the best performance on downstream tasks.
>
>
> ---
>
>
> We have added the following  to algorithm 1 and similarly for algorithm 2:
>
> **Symbols:** $ \theta \in\mathbb{R}^d$ (backbone params), $\mathbf{p}\in\mathbb{R}^{d'}$ (PEFT params), $\mathbf{s}\in\mathbb{R}^{d'}$ (aux inner variable), $G_{\lambda}( \theta ,\mathbf{p},\mathbf{s})$ (penalized objective), $K$ (number of outer steps), $T$ (number of inner steps between two outer steps), $\eta>0$ (inner LR), $\zeta>0$ (outer LR), $\{\lambda_k\}\subseteq \mathbb{R}_+$ (penalty at step $k$).
>
> **Input:** step sizes $\eta,\zeta>0$; initial states $ \theta ^0,\mathbf{p}^0,\mathbf{s}^0$;
>
>
> ---
>
> ## 4- Discussion on other applications
>
> We have added the following paragraph after meta-learning results:
>
> We showed the applicability of bilevel-zofo in Meta Learning. Future work can explore bilevel-zofo in  Multi-Task Reinforcement Learning to tune an LLM on multiple domains. Also another application of bilevel-zofo is in Federated or privacy-sensitive scenarios where clients can run small FO-PEFT steps locally and aggregate ZO signals centrally, which we leave to future work.
>
>
>
> ----
>
> We hope that the major improvements we have made to the text—along with fixes to smaller issues, such as removing forward references from the method to the theory—directly address your remaining concerns. We truly appreciate your feedback as it significantly improved the clarity of our work.

---

> > ### Author Response · Authors · 2025-08-08
> >
> > Dear reviewer WksX,
> >
> > As the discuss period ends today, we’d appreciate knowing whether our response has fully addressed your concerns or if there’s anything that still needs clarification.
> > Thank you again for your time and feedback. It’s greatly appreciated.

---

### Official Review · Reviewer_CENo · 2025-07-02

**Clarity:** 2
**Significance:** 2
**Originality:** 3
**Rating:** 5
**Confidence:** 4

**Summary:**

The paper introduces Bilevel ZOFO, a bi-level fine-tuning algorithm for memory efficient training of LLMs. The method proposed uses improvements from both first-order (FO) PEFT and zeroth-order (ZO) tuning methods to improve overall model quality. The authors identify that ZO take long training durations to converge and have affinity to certain prompt formats. On the other hand, PEFT methods like lora are not able to fully achieve the same model quality as full-finetuning. The authors propose a bilevel optimization framework, where both FO-PEFT + ZO are combined. The base model is updated only using ZO, while the PEFT weights are updated in a low-level pass at some update frequency. By combining these two methods, the authors show better performance over either method on some downstream tasks. They further extend their method to meta-learning and show improvements over conventional methods such as Meta-ICL, while being more compute (memory + throughtput) efficient.

**Questions:**

1. the paper looks at very simple classification tasks for experimental evaluation. Have the authors tried it on more difficult tasks such as code / math tuning and have been able to see any success from the method ?
2. For methods like lora, there have been many more advancements in the literature, that enable high quality PEFT training. Did the authors look at those and see if any of them are more useful in the setting of ZOFO?

**Ethical Concerns:**

["NO or VERY MINOR ethics concerns only"]

**Final Justification:**

The authors addressed my concerns. I did not raise the score beyond the original rating, since the paper is not a groundbreaking work - there are some technical challenges in scaling this work to larger datasets and model sizes.

**Limitations:**

yes

**Quality:**

3

**Strengths And Weaknesses:**

**Strengths**

1. The method has good convergence guarantees and all proposed lemmas have proofs that can be followed.
2. The method improves over standard PEFT and Zero-Order (ZO) methods, including some limitations of ZO methods such as affinity to prompt structure and longer training intervals.
3. The paper has ablations for different design choices related to the final algorithm, showcasing the strengths of the designed method.
4. The method is easily adaptable to meta-learning settings, where it outperforms methods such as Meta-ICL on several tasks, while being more compute efficient and taking lower time per training iteration.

**Weaknesses**

1. For most of the main experimental results, we see comparisons against lora and other peft methods, but there is no full fine-tuning with first order methods to compare as a baseline, so as to understand the gap that ZO or PEFT methods have and how much of that is ZOFO able to cover.
2. There are some parts of the paper which are tricky to follow. For example, in Section 3, when the original method is being introduced, a lot of variables such as $D_f$ and $s$ are explained after their use in equations. Also, usage of dimensions such as d vs d' etc. are not explained beforehand. It took some time and multiple re-reads of the section to fully understand this part. Would recommend rewording and introducing necessary notations for clarity.

---

> ### Author Rebuttal · Authors · 2025-07-31
>
> We thank the reviewer for the positive assessment and the thoughtful feedback. Below we address the main points and list concrete updates made to the paper.
>
> ---
>
> ## 1. Full First-Order Fine-Tuning Baselines
>
>  Our problem setting targets memory/throughput-constrained fine-tuning where full first-order (FO) updates on all backbone parameters are often impractical. This is stated in the motivation (Introduction and Problem Setup). We therefore emphasized comparisons against PEFT and ZO methods that are actually deployable under such constraints.
>
> Per the request, we now include FO-FT results to contextualize the gap. These runs use the same data splits and evaluation protocol as our single-task experiments (see Experimental Setup). We will add a short resource note beside the table so readers can see the cost profile of FO-FT versus PEFT/ZOFO. The resource usage comparison of FO FT and FO PEFT and MeZO can also be found in the MeZO paper.
>
> **OPT-1.3B (FO-FT)**
>
> | boolq | cb    | copa  | record | rte   | sst2  | wic   | winogrande | wsc   |
> |-------|-------|-------|--------|-------|-------|-------|------------|-------|
> | 0.660 | 0.821 | 0.730 | 0.719  | 0.690 | 0.937 | 0.586 | 0.526      | 0.635 |
>
> **Llama-2-7B (FO-FT)**
>
> | boolq | record | squad | sst2  |
> |-------|--------|-------|-------|
> | 0.863 | 0.814  | 0.801 | 0.952 |
>
> The new results do not change our conclusions: **Bilevel-ZOFO outperforms FO-PEFT and MeZO at comparable memory and narrows the gap to FO-FT while remaining much more efficient. It also acts as a strong efficient meta learning algorithm.**
>
> ---
>
> ## 2. Clarity of Section 3 and Notation
>
> We revised  section 3 for readability:
>
> - Introduced parameters and dimensions before use.
> - Reordered equations so definitions come before usage.
> - Added a compact Notation section.
> - Added forward references from main-text claims to the specific lemmas in the appendix.
>
> ---
>
> ## 3. Beyond “Simple” Tasks: Code and Math
>
> We followed MeZO’s protocol for comparability, and these kinds of tasks remain widely used in practice. We agree that adding more challenging tasks Weill improve the paper so we have added harder settings to test generality.
>
> **Setup (Qwen2-7B):**
> GSM8K (train/test), HumanEval (train/test), Math500 (4:1 train/test split).
> LoRA and Bilevel-ZOFO trained for 2,000 steps; MeZO for 10,000 steps.
> Metrics: accuracy for GSM8K/Math500; pass@1 for HumanEval.
>
>
> | Method             | GSM8K | Math500 | HumanEval (pass@1) |
> |--------------------|-------|---------|--------------------|
> | Before tuning      | 0.420 | 0.18    | 0.476              |
> | LoRA               | 0.727 | 0.28    | 0.518              |
> | MeZO               | 0.329 | 0.05    | 0.110              |
> | Bilevel-ZOFO (ours)| 0.762 | 0.31    | 0.543              |
>
> Across all three tasks, Bilevel-ZOFO improves over LoRA and strongly over MeZO.
>
> ---
>
> ## 4. Other PEFT Variants
>
> As we mention in the paper Bilevel-ZOFO is a framework: the lower level can use any FO-PEFT and the upper level any ZO estimator. Another reviewer has mentioned DePT [1] (not a lora variant but a prompt tuning variant) and to demonstrate compatibility with other PEFT methods we added support for that:
>
> **Llama-2-7B with DePT**
>
> | Method           | BoolQ | SST-2 |
> |------------------|-------|-------|
> | DePT             | 0.813 | 0.932 |
> | MeZO             | 0.792 | 0.865 |
> | Bilevel-ZOFO+DePT| 0.852 | 0.946 |
>
> We can see that gains persist when swapping in a stronger PEFT variant. If there is a specific LoRA variant that the reviewer is interested in, we would be happy to compare with it.
>
> ---
> ## 5. References
> [1] DePT: Decomposed Prompt Tuning for Parameter-Efficient Fine-tuning. In ICLR 2024.
>
> ---
>
> Once again we thank the reviewer for their valuable feedback.

---

> > ### Comment · Reviewer_CENo · 2025-08-05
> >
> > I thank the authors for the responses, I have no further questions.

---

> > > ### Author Response · Authors · 2025-08-05
> > >
> > > We thank you for your response and appreciate the feedback you provided. It truly improved our work.

---

> ### Comment · Area_Chair_muWV · 2025-08-05
> **Respond to the authors**
>
> Hi reviewer CENo:
>
> Thanks for your review. Please take some time to see if the authors have addressed your question. Thank you!
>
> Your AC

---

### Decision · Program_Chairs · 2025-09-17

**Decision:**

Accept (poster)

**Comment:**

This paper introduces Bilevel-ZOFO, a fine-tuning framework that integrates zeroth-order (ZO) optimization on the full backbone with first-order (FO) parameter-efficient fine-tuning (PEFT) in a bilevel structure. The FO inner loop stabilizes and accelerates the ZO outer loop, mitigating prompt sensitivity and slow convergence issues observed in pure ZO approaches. The authors provide convergence analysis, ablation studies, and experiments across classification, meta-learning, and reasoning benchmarks (including math and code). Results show that Bilevel-ZOFO improves over LoRA, MeZO, and related baselines, with competitive memory efficiency.

Overall, the majority of reviewers find the paper technically solid and novel enough for NeurIPS. The bilevel formulation is well-motivated, convergence guarantees are provided, and empirical evidences, especially with new results on GSM8K, Math500, and HumanEval, demonstrates the usefulness of the approach beyond small classification tasks.